# The basic helix-loop-helix transcription factor SHARP1 is an oncogenic driver in MLL-AF6 acute myelogenous leukemia

Akihiko Numata[1], Hui Si Kwok[1], Akira Kawasaki[1], Jia Li[1], Qi-Ling Zhou[1], Jon Kerry[2], Touati Benoukraf[1], Deepak Bararia[1], Feng Li[1], Erica Ballabio[2], Marta Tapia[2], Aniruddha J. Deshpande[3], Robert S. Welner[4], Ruud Delwel[5], Henry Yang[1], Thomas A. Milne [2], Reshma Taneja [6] & Daniel G. Tenen [1,7]

Acute Myeloid Leukemia (AML) with *MLL* gene rearrangements demonstrate unique gene expression profiles driven by MLL-fusion proteins. Here, we identify the circadian clock transcription factor *SHARP1* as a novel oncogenic target in MLL-AF6 AML, which has the worst prognosis among all subtypes of *MLL*-rearranged AMLs. SHARP1 is expressed solely in MLL-AF6 AML, and its expression is regulated directly by MLL-AF6/DOT1L. Suppression of SHARP1 induces robust apoptosis of human MLL-AF6 AML cells. Genetic deletion in mice delays the development of leukemia and attenuated leukemia-initiating potential, while sparing normal hematopoiesis. Mechanistically, SHARP1 binds to transcriptionally active chromatin across the genome and activates genes critical for cell survival as well as key oncogenic targets of MLL-AF6. Our findings demonstrate the unique oncogenic role for SHARP1 in MLL-AF6 AML.

[1] Cancer Science Institute of Singapore, National University of Singapore, Singapore 117599, Singapore. [2] MRC Molecular Haematology Unit, MRC Weatherall Institute of Molecular Medicine, NIHR Oxford Biomedical Research Centre Programme, Radcliffe Department of Medicine, University of Oxford, Oxford OX3 9DS, UK. [3] Sanford Burnham Prebys Medical Discovery Institute, La Jolla, CA 92135, USA. [4] Division of Hematology/Oncology, The University of Alabama at Birmingham, Comprehensive Cancer Center, Birmingham, AL 35294, USA. [5] Department of Hematology, Erasmus University Medical Center, 3015 GE Rotterdam, The Netherlands. [6] Department of Physiology, Yong Loo Lin School of Medicine, National University of Singapore, Singapore 117593, Singapore. [7] Harvard Stem Cell Institute, Harvard Medical School, Boston, MA 02115, USA. These authors contributed equally: Akihiko Numata and Hui Si Kwok. Correspondence and requests for materials should be addressed to R.T. (email: phsrt@nus.edu.sg) or to D.G.T. (email: daniel.tenen@nus.edu.sg)

The *MLL* (mixed lineage leukemia) gene is located on chromosome *11q23* and encodes a large histone methyl-transferase. MLL constitutes a large protein complex, binding to DNA and positively regulates the clustered homeobox (*HOX*) genes through histone 3 lysine 4 (H3K4) methyl-transferase activity of the SET domain[1,2] and histone acetyl-transferase activity of p300/CBP, MOZ, and MOF interacting with the PHD or TA domain[3–5]. The translocation of *11q23* is one of the most frequent chromosomal abnormalities in acute leukemia, and this rearrangement fuses the genomic region encoding the N-terminus of *MLL* to a sequence encoding the C-terminus of one of a number of fusion partner proteins, resulting in loss of chromatin modification potential. MLL-fusion protein (MLL-FP) acquires a unique transcriptional machinery recruiting the transcriptional elongation complex, EAP (elongation assisting protein), that includes p-TEFb (positive transcription elongation factor b), which phosphorylates RNA polymerase 2 and results in sustained transcriptional elongation[6]. The MLL-FP also interacts with DOT1L (disruptor of telomeric silencing 1-like), a specific H3K79 methyltransferase; di- and tri-methylated H3K79 (H3K79me2/3) are epigenetic hallmarks of active transcription by MLL-FPs[7]. Pharmacological inhibition or genetic deletion of DOT1L substantially suppresses *MLL*-rearranged (MLLr) AML[8,9], indicating it as a therapeutic target.

More than 70 genes have been characterized as partner genes of *MLL* in acute leukemia[10]. Although the partner proteins have various functions and cellular localizations, most of the MLL-FPs share a principle machinery in their transcriptional regulation. AF4, AF9, AF10, and ENL are nuclear partner proteins that form a part of the transcriptional elongation complex, and these fusion partners account for more than 80% of all clinical cases of MLLr acute leukemias[10]. On the other hand, MLL-AF6 represents the most common leukemogenic fusion of MLL to a cytoplasmic partner protein. AF6 is not identified in the components of the major transcriptional elongation complex[7,11]. Nevertheless, MLL-AF6 also recruits EAP and DOT1L complexes to target chromatin via an unknown mechanism and activates transcriptional elongation of target genes[7,12] and the unique underlying mechanisms for MLL-AF6-driven leukemogenesis have not been fully eluci-dated. Here, we identify a basic helix-loop-helix transcription factor *SHARP1* as a MLL-AF6 specific target gene and revealed its unique oncogenic role, representing a potential therapeutic target.

## Results

**SHARP1 is overexpressed in MLL-AF6 AML**. To uncover specific underlying mechanisms for MLL-AF6 AML, we identified direct transcriptional target genes of MLL-AF6. To this end, we performed chromatin immunoprecipitation followed by deep sequencing (ChIP-seq) using the ML-2 cell line, which is derived from a patient with AML harboring t(6;11)(q27;q23) and lacks endogenous full-length *MLL* gene[13,14]. The N-terminus of MLL (MLL[N]), when fused to its fusion partners, recruits the H3K79 methyltransferase DOT1L directly or indirectly, and methylation of H3K79 was linked to active transcribed MLL-AF6 target genes[12]. Thus the use of antibodies against MLL[N] and dimethy-lated H3K79 (H3K79me2) enabled us to identify actively tran-scribed MLL-AF6 target genes. We identified 92 genes showing overlap of MLL[N] (101 genes) (Supplementary Tables 1 and 2) and H3K79me2 (8904 genes) peaks in their gene loci, which are potentially regulated by MLL-AF6 (Fig. 1a). This gene set includes the posterior *HOXA* genes (*HOXA7, 9, 10*), *JMJD1C*, *MEF2C*, and *MYB*, which were identified as target genes of MLL-FPs in previous studies[15–18]. To identify specific targets of MLL-AF6, we further interrogated gene expression profiles of adult AML patients, comparing MLL-AF6 (14 cases) to the other

subtypes of MLLr-AML (42 cases) and found 581 genes sig-nificantly upregulated in MLL-AF6 AML patients (Log2 fold > 0.5, $p < 0.05$). Among these genes, we identified nine MLL-AF6 targets (*SHARP1, P2RY1, SSPN, FAM169A, TRPS1, MMRN1, SKIDA1, HOXA7*, and *SLC35D1*) (Fig. 1b, c, Table 1, and Sup-plementary Fig. 1a), whereas there was no difference in the expression level of MLL-FPs canonical targets (*HOXA9, HOXA10*, and *MEIS1*) between MLL-AF6 and the specific sub-types generally (Supplementary Fig. 1b).

Basic helix-loop-helix (bHLH) transcription factor *SHARP1* (also known as *BHLHE41* or *DEC2*) was the highest and the most significantly upregulated MLL-AF6 target gene (average log2 fold change 4.650, $-\log_{10} p$ value 13.32) (Fig. 1b and Table 1). Although *SHARP1* was identified as a common retroviral integration site in the genomes of AKXD murine myeloid tumors[19], suggesting a potential role in leukemogenesis, there have not been further studies on its role in leukemogenesis. Importantly, SHARP1 was decreased in most cases of other subtypes of AML as well as normal bone marrow (NBM) CD34[+] cells (Fig. 1c). Moreover, to test these findings, unsupervised hierarchical gene-expression clustering of leukemic blasts of adult AML patients from two independent cohorts was performed. Three cases, in a cohort of 285 AML cases that were studied using gene expression profiling, showed high SHARP1 expression levels (Fig. 1d). These three cases were in a cluster that was highly enriched for AMLs with a MLL-rearrangement (MLLr-AML)[20] and all three carried a t(6;11). Gene expression profiling of a second cohort of AMLs ($n = 268$) revealed two more cases with high SHARP1 expression, which also carried a t(6;11), and were clustered within a group of patients with MLLr-AML as well (Fig. 1e). In these two cohorts, all of the MLL-AF6 AML cases showed high SHARP1 expression. These findings prompted us to investigate whether SHARP1 plays an important role in the pathogenesis of MLL-AF6 AML.

**MLL-AF6 directly upregulates SHARP1 by DOT1L**. In human AML cell lines, consistent with our findings in the gene expres-sion profiles from the multiple AML cohorts, SHARP1 mRNA was expressed highly in ML-2, CTS and SHI-1 cells, all of which harbor t(6;11)(q27;q23), whereas it was undetectable in MOLM-14, MV4-11 and Kasumi-1, which harbor t(9;11)(p22;q23), t(4;11)(q21;q23), and t(8;21)(q22;q22), respectively (Fig. 2a). MLL-FP complex contains MEN1 (Multiple Endocrine Neoplasia syndrome type 1, also called MENIN), which binds to the N-terminus of MLL, linking it to LEDGF (Lens Epithelium-Derived Growth Factor). The association of MEN1 or LEDGF with MLL is required for chromatin localization of the complex and tran-scription of their target genes, which are crucial for MLLr-leukemias development[18,21]. A histone methyltransferase, DOT1L is a subunit of MLL-FP complexes and solely responsible for both H3K79 di- and tri-methylation (H3K79me2/3). ChIP-seq analysis of ML-2 cells demonstrated that posterior *HOXA* genes (*HOXA7-10*) were bound by MLL[N]/MEN1/LEDGF and enriched with H3K79me2/3, which is a hallmark of DOT1L recruitment to active chromatin, whereas the region of anterior *HOXA* genes (*HOXA1-6*) were neither bound by MLL[N]/MEN1/LEDGF nor enriched with H3K79me2/3 (Fig. 2b). In the *SHARP1* gene locus, MLL[N]/MEN1/LEDGF localized across the transcribed region concomitantly with high enrichment of H3K79me2/3 (Fig. 2b). These findings were verified by ChIP-quantitative PCR (qPCR) of the promoter regions of the *SHARP1* gene using antibodies against MLL[N] and H3K79me2 and ChIP-qPCR of *HOXA9* pro-moter was used as a positive control (Supplementary Fig. 2a). To confirm these findings in another MLL-AF6 AML cell line, we performed an independent ChIP-seq analysis of SHI-1 cells which

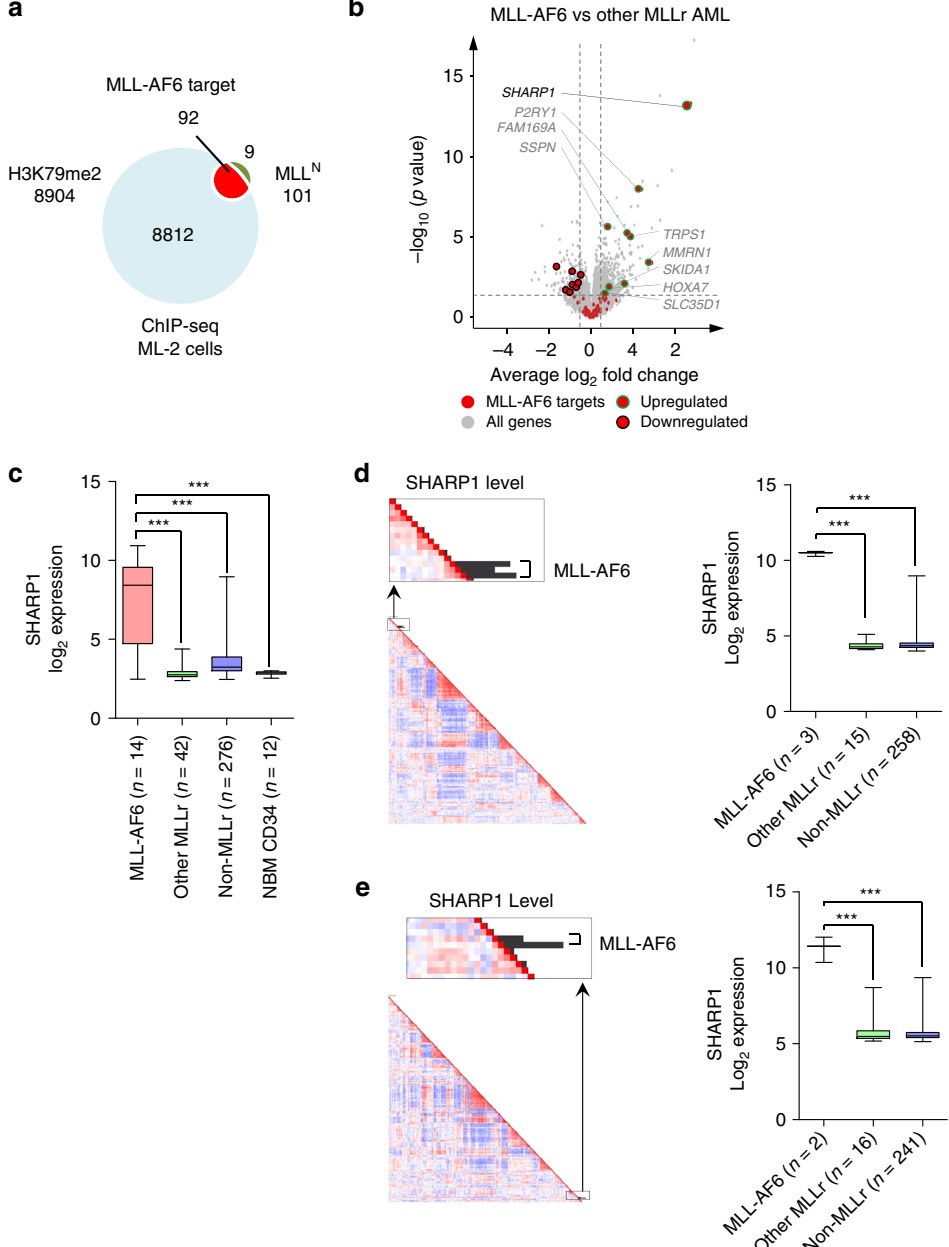

**Fig. 1** Overexpression of *SHARP1* in MLL-AF6 AML patients. **a** Venn diagram showing MLL-bound (101 genes) and H3K79me2 enriched genes (8904 genes) obtained from ChIP-seq analysis of ML-2 cells for identification of 92 MLL-AF6 target genes. **b** Volcano plot showing average log$_2$ fold change against −log$_{10}$ p value for all genes in MLL-AF6 AML ($n = 14$) vs all the other subtypes of *MLL*-rearranged AMLs (other MLLr) ($n = 42$). Gene expression data of patients were obtained from GSE19577, GSE14468 and GSE61804. Red dots MLL-AF6 target (92 genes), Green circles upregulated targets (9 genes), Black circles downregulated targets (8 genes). **c** Box plot showing SHARP1 log$_2$ expression in AML patients and normal bone marrow (NBM) CD34 + cells. SHARP1 log$_2$ expression level: MLL-AF6 7.504 ± 0.788 ($n = 14$), Other MLL 2.854 ± 0.065 ($n = 42$), Non-MLL 3.623 ± 0.064 ($n = 276$), NBM CD34 2.856 ± 0.036 ($n = 12$). Gene expression data of NBM CD34$^+$ cells were from GSE19429. **d, e** Left panel: Unsupervised hierarchical gene-expression clustering from 2 distinct cohorts of adult AML patients from GSE1159 (**d** $n = 285$) and GSE6891 (**e** $n = 268$). The bars indicate SHARP1 gene expression. All of the five high SHARP1 cases have the MLL-AF6 fusion gene. Right panel: Box plot showing SHARP1 log$_2$ expression in AML patients. SHARP1 log$_2$ expression level: **d** MLL-AF6:10.45 ± 0.096 ($n = 3$), Other MLL 4.383 ± 0.082 ($n = 15$), Non-MLL 4.566 ± 0.045 ($n = 258$). The cytogenetics was not determined in 9 cases. **e** MLL-AF6 11.18 ± 0.828 ($n = 2$), Other MLL 5.916 ± 0.257 ($n = 16$), Non-MLL 5.658 ± 0.033 ($n = 241$). The cytogenetics was not determined in 9 cases. All box plots extend from the 25$^{th}$ to 75$^{th}$ percentiles and the whisker extends from the minimum level to the maximum. Median value is plotted in the box. ***$p < 0.001$

expresses both MLL and MLL-AF6, demonstrating that MLL$^N$ binds to *SHARP1* gene loci, as well as posterior *HOXA* genes locus (Fig. 2c). To ascertain the unique MLL-AF6 binding, we analyzed MLL$^N$ and H3K79me2 ChIP-seq data of THP-1 (MLL-AF9) and MV4-11 (MLL-AF4) cells and found that neither MLL$^N$

binding nor H3K79me2 enrichment was observed at *SHARP1* loci (Supplementary Fig. 2b). Collectively, our results indicate that SHARP1 is a unique transcriptional target of MLL-AF6 and its expression is not suppressed at the post-transcriptional level in the other MLLr-AML subtypes.

### Table 1 MLL-AF6 specific target genes

| Gene | Description | Fold change(log2) | p value(-log10) |
|------|-------------|-------------------|-----------------|
| *SHARP1* | Basic helix-loop-helix family, member e41 | 4.65 | 13.3 |
| *P2RY1* | Purinergic receptor P2Y1 | 2.35 | 8.00 |
| *SSPN* | Sarcospan | 0.81 | 5.65 |
| *FAM169A* | Family with sequence similarity 169, member A | 1.71 | 5.21 |
| *TRPS1* | Trichorhinophalangeal syndrome 1 | 1.89 | 5.02 |
| *MMRN1* | Multimerin 1 | 2.82 | 3.43 |
| *SKIDA1* | SKI/DACH domain containing 1 | 1.59 | 2.13 |
| *HOXA7* | Homeobox A7 | 0.91 | 1.89 |
| *SLC35D1* | Solute carrier family 35, member D1 | 0.58 | 1.39 |

A list of nine MLL-AF6 target genes presenting higher expression in MLL-AF6 AML than all the other subtypes of MLLr-AMLs

To examine whether MLL-AF6 regulates SHARP1 expression, we performed MLL-AF6 knockdown using two independent lentiviral shRNA targeting MLL$^N$ (shMLL #1 and #2) in ML-2 cells. Reduction in MLL-AF6 resulted in suppressed SHARP1 mRNA expression (Fig. 2d). Pharmacological inhibition of DOT1L results in robust and selective ablation of H3K79 methylation, leading to suppressed transcription of MLL-FP target genes, such as *HOXA* gene cluster and *MEIS1*[9]. To investigate whether SHARP1 expression relies on DOT1L activity, ML-2 and MOLM-14 cells were treated with the selective aminonucleoside inhibitor EPZ5676[22]. Consistent with the previous study, H3K79me2 was reduced in both cell lines (Fig. 2e). Importantly, EPZ5676 treatment dramatically reduced SHARP1 expression at both mRNA and protein levels in ML-2 cells, whereas no significant change was observed in MOLM-14 cells at mRNA level (Fig. 2f, g). Collectively, these results demonstrate that MLL-AF6 and MEN1/LEDGF directly bind to the *SHARP1* gene locus to positively regulate its expression through DOT1L activity.

**SHARP1 maintains clonogenic growth of MLL-AF6 AML cells.** To elucidate the role of SHARP1 in human MLL-AF6 leukemia, we performed knockdown experiments using two independent lentiviral shRNAs against SHARP1 (shSHARP1 #1 and #2) and a shRNA against GFP (shGFP) as a control in ML-2, CTS and SHI-1 cells. Knockdown efficiency was confirmed by qPCR and Western Blotting (Fig. 3a and Supplementary Fig. 3a). Equal number of the cells was injected intravenously into sublethally irradiated (240 rads) NOD-SCID common gamma chain deficient (NSG) mice. Recipients of ML-2 or CTS shSHARP1 showed significantly extended survival length than those of shGFP (median survival; ML2 shGFP 41.5, shSHARP1#1 45, shSHARP1#2 58.5 days, shGFP vs shSHARP1#1 $p = 0.0008$, shGFP vs shSHARP1#2 $p = 0.0003$, CTS shGFP 22, shSHARP1#1 25, shSHARP1#2 25.5 days, shGFP vs shSHARP1#1 $p = 0.0185$, shGFP vs shSHARP1#2 $p = 0.0065$) (Fig. 3b). Consistently, downregulation of SHARP1 increased apoptotic cells (AnnexinV$^+$ DAPI$^-$ or PI$^-$) (Fig. 3c), while granulocytic and monocytic differentiation was not observed, assessed by flow cytometry and morphological analysis (Supplementary Fig. 3b). We also observed attenuated cell growth (Fig. 3d) and colony formation (Fig. 3e). However, transduction of the two SHARP1 shRNA neither induced apoptosis nor attenuated cell growth and colony-forming ability in MOLM-14 (MLL-AF9) and MV4-11 (MLL-AF4) (Supplementary Figs. 3c to 3e). Collectively, our results demonstrate a critical role of SHARP1 in maintaining clonogenic growth and preventing apoptosis of MLL-AF6 AML cells.

**Deletion of Sharp1 attenuates MLL-AF6 AML progression.** To further investigate the role of SHARP1 in the development of MLL-AF6 AML, we transduced fluorescence-activated cell sorting (FACS)-sorted lineage (Lin)$^-$ Sca1$^+$ c-kit$^+$ (LSK) cells from bone marrow (BM) cells of *Sharp1* $^{+/+}$ and *Sharp1* $^{-/-}$ mice[23] with the MLL-AF6 fusion gene as described previously[12]. A total of 200,000 transduced cells were transplanted into sublethally irradiated (650 rads) congenic mice (Fig. 4a). In the long-term follow up, the recipients of MA6/S1KO demonstrated significantly longer survival than those of MA6/WT (median survival; MA6/WT 111.5 vs MA6/S1KO 77 days, $p = 0.0002$) (Fig. 4b). Interestingly, peripheral blood (PB) taken 2 months after the transplantation revealed that 14 out of 17 recipients of MA6/WT presented with AML cells (CD45.2$^+$ CD11b$^+$) higher than 20 % of all nucleated cells, as compared to only 5 out of 17 MA6/KO recipients. Recipients of MA6/WT demonstrated higher white blood cell (WBC) counts (median 26.5 vs 7.18 × 10$^3$/μL, $p < 0.001$) and lower red blood cell (RBC) counts (median 6.51 vs 8.74 × 10$^6$/μL, $p < 0.01$), as compared to MA6/S1KO (Fig. 4c), suggesting that *Sharp1* deletion decreased disease aggressiveness. Moribund recipients from both groups displayed liver and spleen enlargement (Fig. 4d). The majority of the BM cells were immature Gr1$^+$ CD11b$^+$ myeloblasts (Fig. 4d, e and Supplementary Fig. 4a) and had similar differentiation status between the two groups (Fig. 4e). To assess the propagative ability of the leukemia cells, 200,000 whole BM cells from leukemic mice were injected into sublethally irradiated (650 rads) congenic mice (Fig. 4a). Recipients of MLL-AF6 AML *Sharp1*$^{-/-}$ presented significantly longer survival than those of *Sharp1*$^{+/+}$ (median survival; 25 vs 17 days, $p < 0.0001$) (Fig. 4b). Consistent with these findings, the colony-forming replating assay, commonly used as a surrogate for assessing leukemic transformation, demonstrated fewer numbers of colonies from the second plating of *Sharp1*$^{-/-}$ cells compared to *Sharp1*$^{+/+}$ (Supplementary Fig. 4b). Collectively, these findings demonstrate that *Sharp1* contributes to the development and propagation of MLL-AF6 AML.

To investigate whether *Sharp1* deletion affects the initiation of other subtypes of MLLr-AML, the MLL-AF9 fusion gene[15] was retrovirally transduced into LSK cells from *Sharp1*$^{+/+}$ or *Sharp1*$^{-/-}$ mice and subsequently 200,000 cells were transplanted into sublethally irradiated (650 rads) CD45.1$^+$ congenic mice (Supplementary Fig. 4c). Recipients from both groups succumbed to leukemia with a similar median survival in primary transplantations (median survival; 74 vs 70 days, $p = 0.302$). Secondary transplantation was performed by injecting 200,000 leukemic whole BM cells into sublethally irradiated (650 rads) congenic mice, which did not exhibit any survival difference (median survival; 23.5 vs 23 days, $p = 0.848$) (Supplementary Fig. 4d), demonstrating that *Sharp1* deletion does not affect development or propagation of MLL-AF9 AML. Consistent with these findings, Sharp1 mRNA was elevated in MLL-AF6 AML cells compared to BM Granulocyte-Macrophage Progenitor (GMP) and granulocytes, which are the phenotypic normal

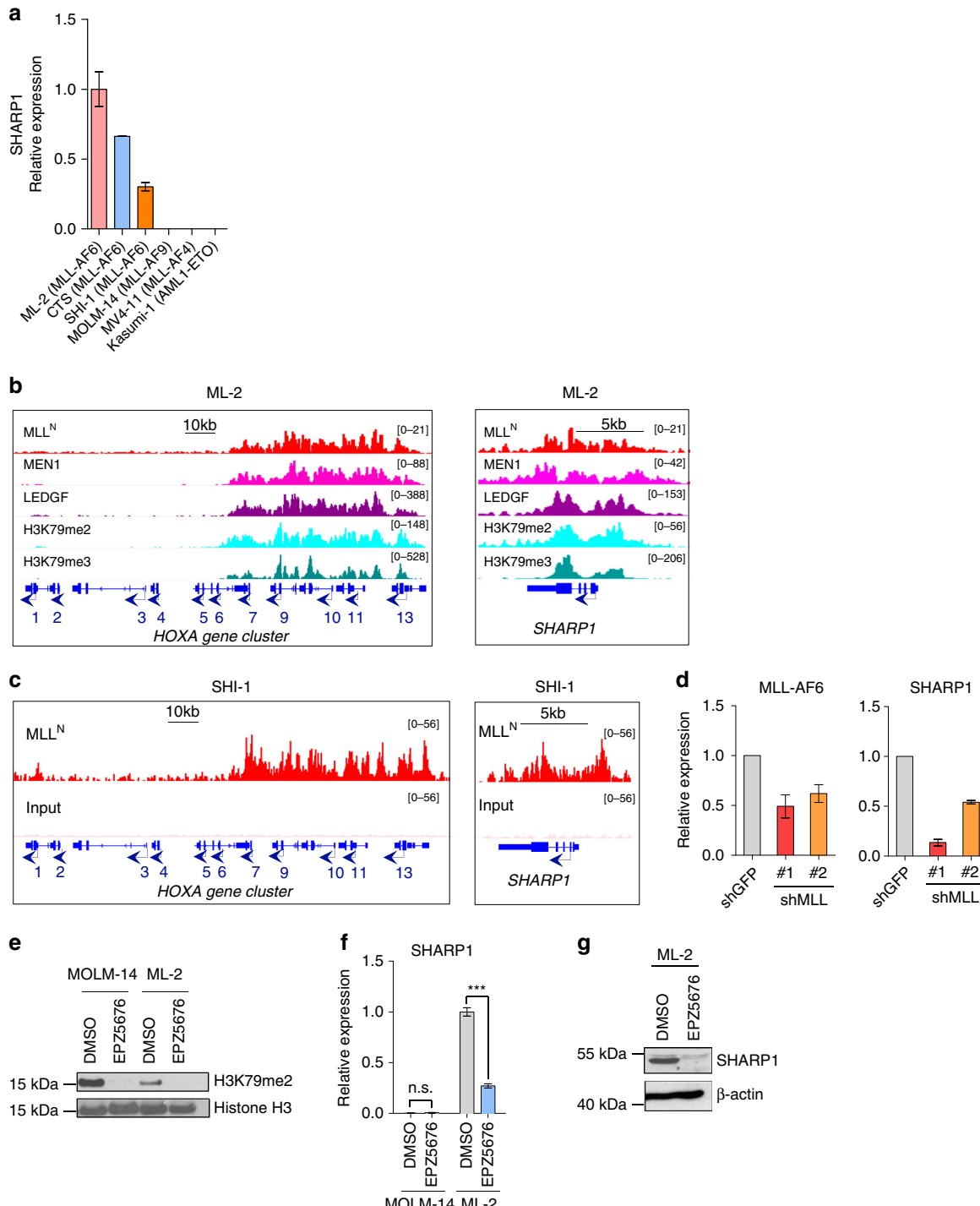

**Fig. 2** *SHARP1* is a downstream target of MLL-AF6 and DOT1L. **a** SHARP1 mRNA expression in human AML cell lines assessed by qPCR. The cell lines analyzed are: ML-2 (MLL-AF6), CTS (MLL-AF6), SHI-1 (MLL-AF6), MOLM-14 (MLL-AF9), MV4-11 (MLL-AF4), and Kasumi-1 (AML1-ETO). **b** ChIP-seq profiles of ML-2 cells using MLL[N], MEN1, LEDGF, H3K79me2, and H3K79me3 antibodies at the loci of the *HOXA* gene cluster (left panel) and *SHARP1* gene (right panel). **c** ChIP-seq profiles of SHI-1 cells using MLL[N] antibody and input at the loci of the *HOXA* gene cluster (left) and *SHARP1* gene (right). **d** qPCR for MLL-AF6 (left panel) and SHARP1 (right panel) mRNA expression in ML-2 cells upon MLL knockdown. Shown is the relative expression value to ML-2 transduced with shGFP. **e** Western blot of H3K79me2 in MOLM-14 (MLL-AF9) and ML-2 (MLL-AF6) cells treated with the DOT1L inhibitor, EPZ5679 (1 μM) or DMSO vehicle for 96 h. **f** qPCR for SHARP1 mRNA in MOLM-14 or ML-2 cells treated with EPZ5679 or DMSO vehicle for 6 days. Relative expression is the value compared to ML-2 cells treated with DMSO vehicle. **g** Western blot of SHARP1 and β-actin in ML-2 cells treated with EPZ5679 or DMSO vehicle for 10 days. All Western blots are representative of three independent experiments. All quantitation data include three independent experiments and are presented as mean ± s.e.m

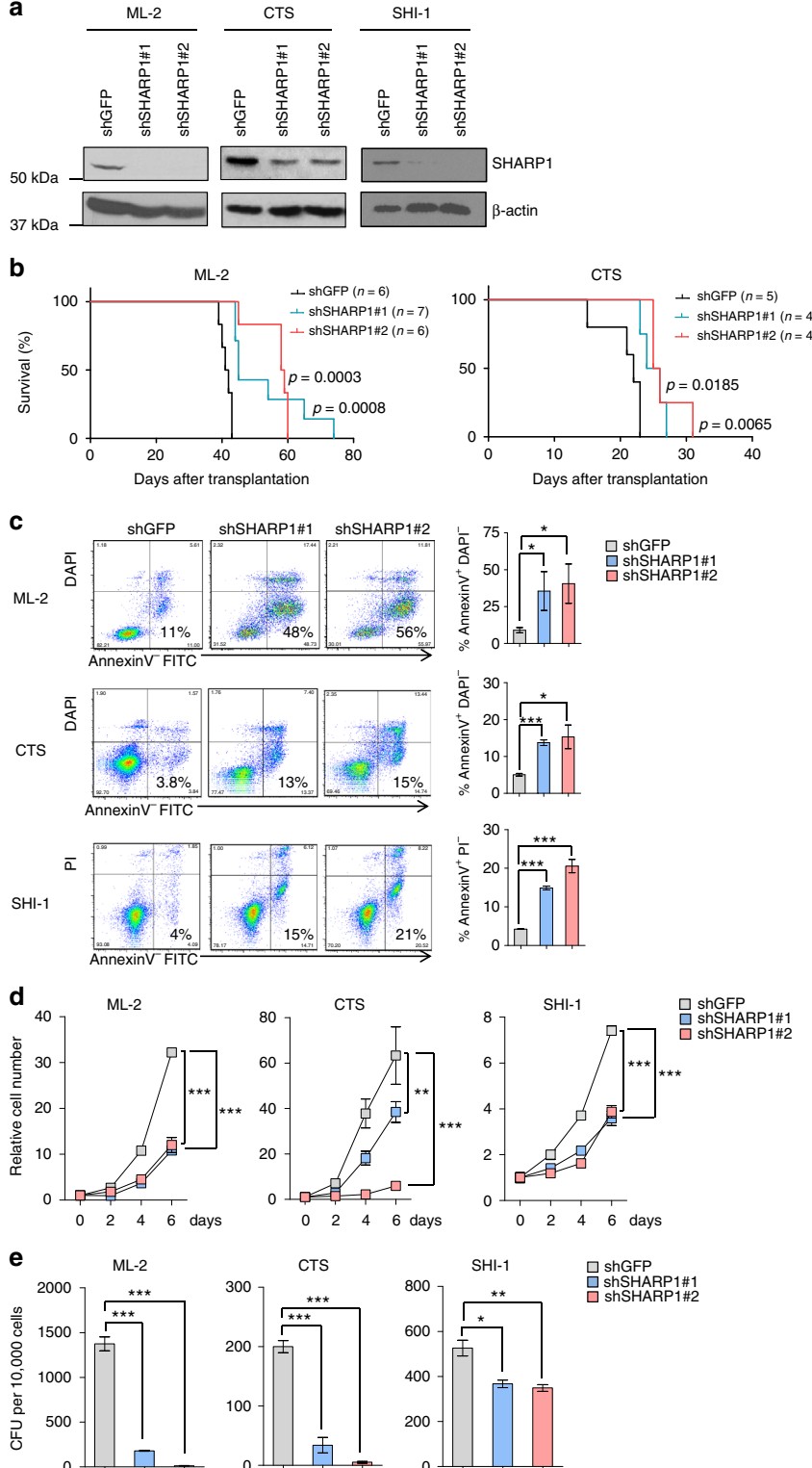

**Fig. 3** SHARP1 is crucial for clonogenic growth of human MLL-AF6 AML cells. **a** Western blots of SHARP1 protein in ML-2, CTS and SHI-1 cells transduced with the indicated shRNAs. Western blots are representative of at least three independent experiments. **b** Kaplan–Meyer survival curve of sublethally irradiated recipient NSG mice transplanted with $5 \times 10^4$ ML-2 or $1 \times 10^5$ CTS cells transduced with the indicated shRNAs. $P$ values are determined by the Log-rank (Mantle-Cox) Test. **c** Left panel: Representative flow cytometry plot of ML-2, CTS, and SHI-1 cells transduced with the indicated shRNAs for AnnexinV and DAPI or PI. Right panel: Percentage of AnnexinV[+] and DAPI[−] or PI[−] cell are shown. **d** Cell count of ML-2, CTS, and SHI-1 cells transduced with the indicated shRNAs in culture. The value is determined as fold increase in cell number relative to the number of cells initially plated. **e** Colony-forming units (CFU) of ML-2, CTS, and SHI-1 cells transduced with the indicated shRNAs. The number of colonies observed 7 days after the plating. The graphs are representative examples of three independent experiments and presented as mean ± s.e.m, $*p < 0.05$, $** \ p < 0.01$, $***p < 0.001$

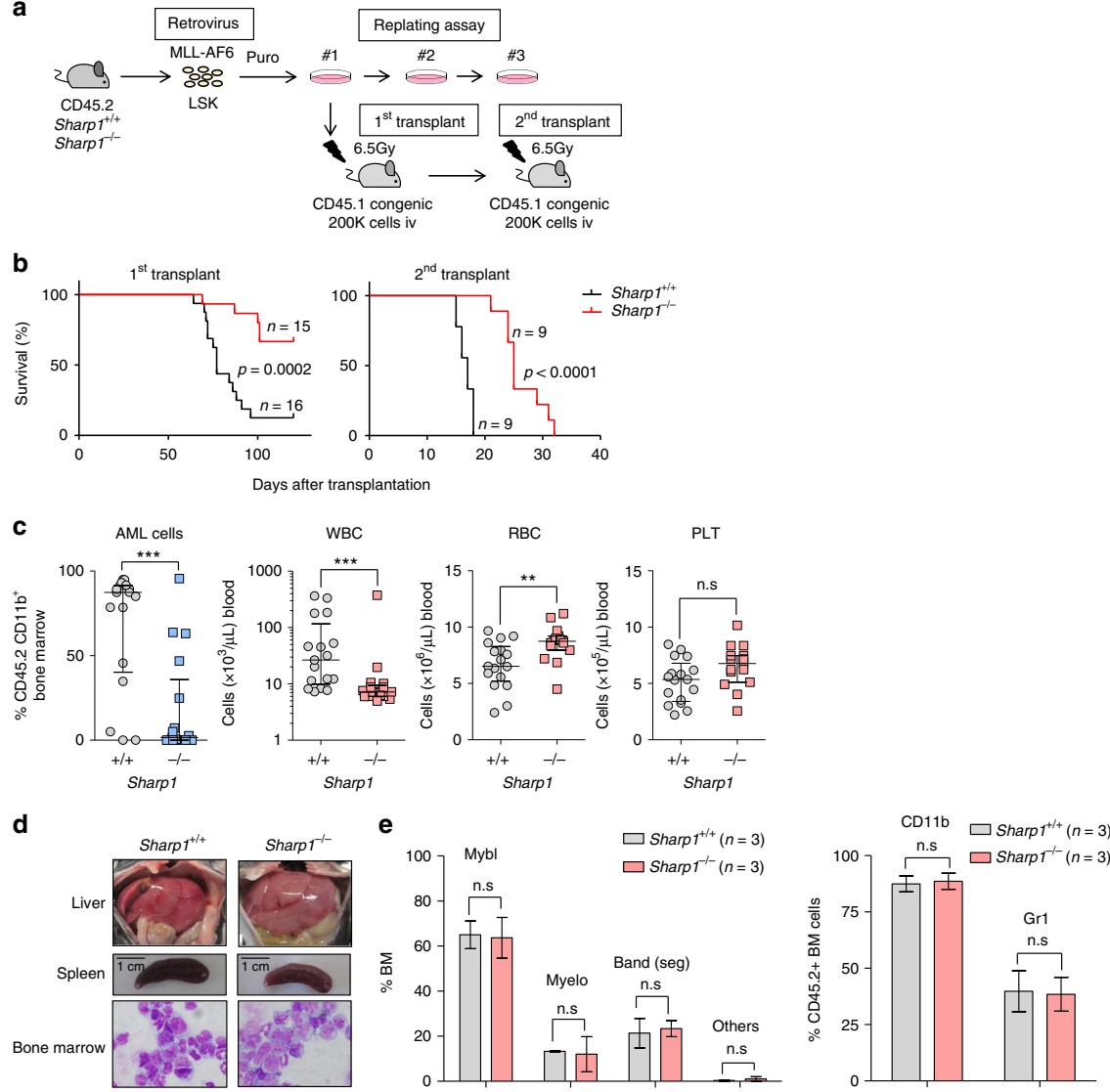

**Fig. 4** Sharp1 deletion attenuates MLL-AF6 AML progression. **a** Experimental strategy for replating and in vivo leukemia assays in Sharp1$^{+/+}$ or Sharp1$^{-/-}$ hematopoietic stem/progenitor cells following retroviral transduction of MLL-AF6. **b** Kaplan–Meyer survival curve of sublethally irradiated congenic mice transplanted with 200,000 cells from (left panel) the first replate (right panel) and whole bone marrow cells isolated from leukemic recipients following the first transplant. Data includes at least two independent transplantation experiments. Statistical analysis was performed using Log-rank (Mantle-Cox) test. **c** Peripheral blood count and percentage of leukemia cells (CD45.2$^{+}$ CD11b$^{+}$) from the recipient mice (n = 17 per genotype) two months after primary transplant are shown. Statistical analysis was performed using Mann–Whitney U test, **p < 0.01, ***p < 0.001. All individual data points include three independent transplantation experiments and are presented as mean ± s.e.m. **d** Representative pictures of liver and spleen and Wright Giemsa staining of bone marrow (BM) cells from moribund leukemic mice. **e** Left panel: Differential count of Wright Giemsa-stained BM cells from moribund leukemic mice. Mybl myeloblasts, Myelo promyelocytes, myelocytes, and metamyelocytes, Band (seg) band and segmented neutrophils, Others lymphocytes and macrophages. Right panel: Percentage of CD11b high and Gr1 high cells in CD45.2$^{+}$ BM cells are shown. The graph is present as mean ± s.e.m from three independent moribund leukemic mice

hematopoietic counterparts for AML cells[15], while the expression in MLL-AF9 AML was comparable to normal BM GMP and higher than granulocytes (Supplementary Fig. 4e), suggesting that upregulated Sharp1 may confer oncogenic properties to murine MLL-AF6 AML, but not to MLL-AF9 AML.

**Sharp1 deletion reduces MLL-AF6 leukemia-initiating ability.** Leukemia cells are heterogeneous and organized as a hierarchy that originates from a small fraction of cells that have self-renewal potential, known as leukemic stem cell (LSC) or leukemia-initiating cell (LIC)[24]. In MLLr-AMLs, MLL-FPs confer stem cell-like properties on committed progenitor and leukemic GMP

(L-GMP) was defined as the cell population enriched for LSC in murine MLL-AF9 AML[15]. To investigate the role of Sharp1 in leukemia-initiating potential, we first assessed the frequency of L-GMP (Lin$^{-}$ c-kit$^{+}$ Sca1$^{-}$ CD34$^{+}$ CD16/32$^{+}$) in MLL-AF6 AML cells by flow cytometry. Interestingly, MLL-AF6 AML Sharp1$^{-/-}$ cells demonstrated significant reduction both in L-GMP and LSK populations compared to Sharp1$^{+/+}$ (Sharp1$^{+/+}$ vs Sharp1$^{-/-}$; 0.50 vs 0.18 %, p < 0.05 and 0.77 vs 0.24 %, p < 0.05, respectively) (Fig. 5a). To assess the leukemia-initiating potential, we performed limiting dilution assay (LDA) by injecting sublethally irradiated (650 rads) congenic recipient mice with limiting number of FACS-sorted L-GMP (5, 50, and 500 cells) or whole BM cells (100, 200, and 2000 cells) from both groups.

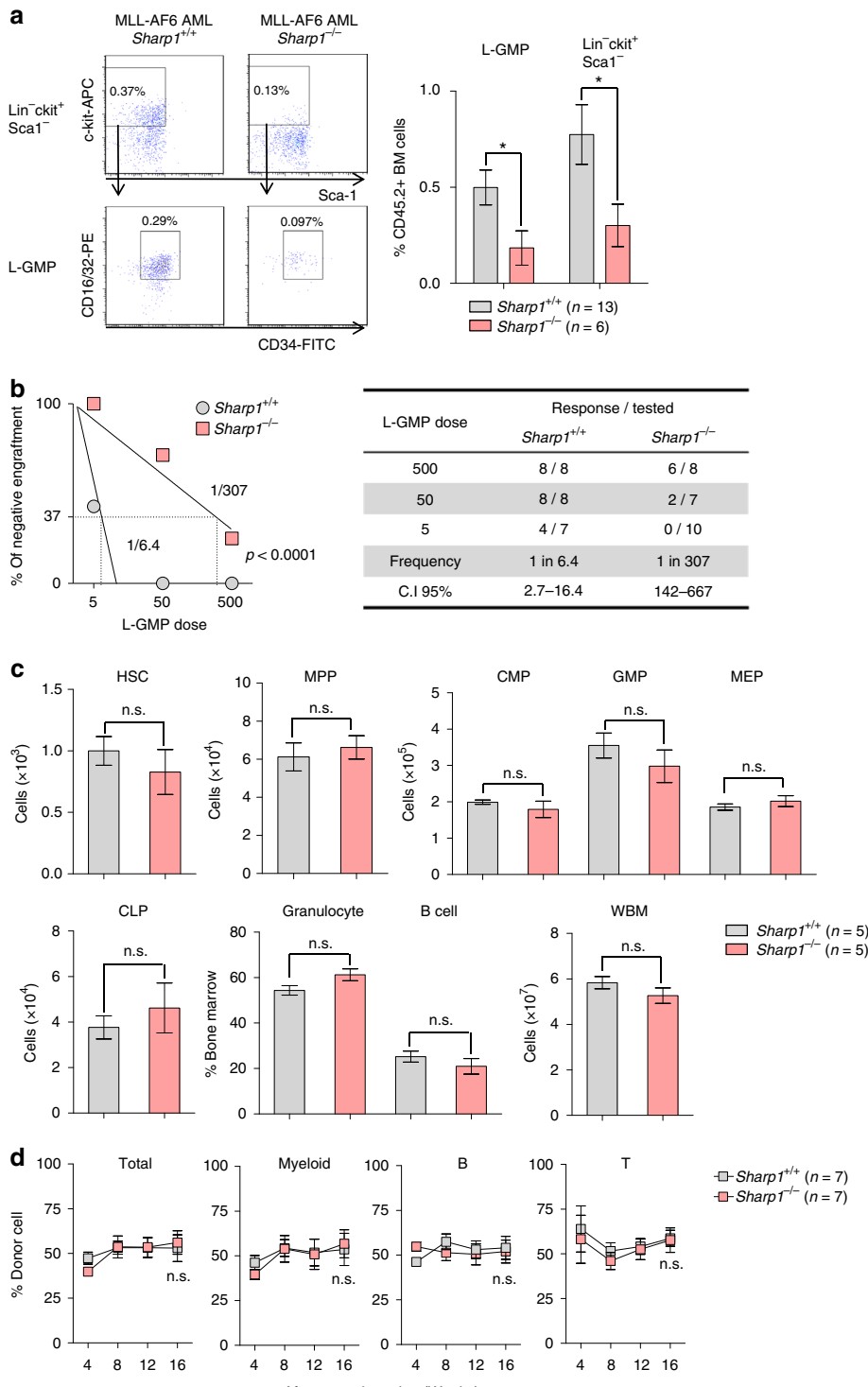

**Fig. 5** Reduced leukemia-initiating cells in $Sharp1^{-/-}$ MLL-AF6 AML. **a** Analysis of Leukemic-Granulocyte Macrophage Progenitor (L-GMP). Left panel: a representative flow cytometry plot, pregated on lineage-negative cells, and then on c-kit$^+$Sca1$^-$CD34$^+$CD16/32$^+$ cells; shown are percentage of total CD45.2$^+$ bone marrow (BM) cells. Percentage of the gated populations were shown. Right panel: Graphical presentation of percentage of L-GMP and Lin$^-$ c-kit$^+$Sca1$^-$ cells in CD45.2$^+$ BM cells. The graph is presented as mean ± s.e.m. **b** Limiting Dilution Assay (LDA). The indicated numbers of FACS-sorted L-GMP (MLL-AF6 AML $Sharp1^{+/+}$ and $Sharp1^{-/-}$) cells were transplanted into sublethally irradiated congenic mice. **c** HSPC and whole bone marrow (WBM) cell number from a femur and a tibia, and percentage of BM mature myeloid cells (Gr1$^+$CD11b$^+$) and B (B220$^+$) are graphed. HSC (hematopoietic stem cell = CD150$^+$CD48$^-$), MPP (multipotent progenitor = CD150$^-$CD48$^+$), GMP (granulocyte-macrophage progenitor = CD34$^+$CD16/32$^+$), CMP (common myeloid progenitor = CD34$^+$CD16/32$^-$), MEP (megakaryocyte-erythroid progenitor = CD34$^-$CD16/32$^-$), CLP (common lymphoid progenitor = IL7R$^+$Flk2$^+$), and WBM. **d** 500,000 WBM cells from $Sharp1^{+/+}$ or $Sharp1^{-/-}$ mice were transplanted into irradiated recipient mice along with 500,000 congenic WBM cells as competitor cells. Percentage of donor in peripheral blood was monitored for 16 weeks after transplantation. The graphs are presented as mean ± s.e.m, *p < 0.05

Remarkably, $Sharp1^{-/-}$ L-GMP harbored dramatically reduced LSC frequency compared to $Sharp1^{+/+}$ ($Sharp1^{+/+}$ vs $Sharp1^{-/-}$; 1:6.4 vs 1:307, $p < 0.001$) (Fig. 5b), whereas $Sharp1^{-/-}$ leukemic BM cells showed 2.3-fold reduction compared to $Sharp1^{+/+}$ ($Sharp1^{+/+}$ vs $Sharp1^{-/-}$; 1:531 vs 1:1,213, $p < 0.05$) (Supplementary Fig. 5a). Consistent with the previous findings in the study of MLL-AF9 AML[15], L-GMP of MLL-AF6 AML cells were markedly enriched for leukemia-initiating potential compared to whole BM (L-GMP vs whole BM; 1:6.4 vs 1:531, $p < 0.001$) (Fig. 5b and Supplementary Fig. 5a). Collectively, $Sharp1$ deletion attenuated leukemia-initiating potential of MLL-AF6 AML cells, and this effect was more profound in the L-GMP population. The prolonged survival in the recipients of MLL-AF6 AML $Sharp1^{-/-}$ in the secondary transplants could be explained by lower numbers of transplanted LSC.

**Sharp1 is dispensable for steady-state hematopoiesis.** Given that $Sharp1$ plays a role in L-GMP maintenance in MLL-AF6 AML, we asked whether $Sharp1$ deletion affects normal hematopoiesis, especially the committed myeloid progenitor cells. We first analyzed steady-state BM cells obtained from sex and age-matched $Sharp1^{+/+}$ and $Sharp1^{-/-}$ mice. We did not find any differences in the number of hematopoietic stem cell (HSC = $CD150^+CD48^-$LSK) and progenitor populations (MPP = multipotent progenitor $CD150^-$ $CD48^+$ LSK; CMP = common myeloid progenitor $Lin^-$ $c$-$kit^+$ $Sca1^-$ $CD34^+$ $CD16/32^-$; GMP = $Lin^-$ $c$-$kit^+$ $Sca1^-$ $CD34^+$ $CD16/32^+$; MEP = myeloid erythroid progenitor $Lin^-$ $c$-$kit^+$ $Sca1^-$ $CD34^-$ $CD16/32^-$; and CLP = common lymphoid progenitor, $Lin^-$ $IL7R^+$ $c$-$kit^+$ $Sca1^+$ $Flk2^+$) between $Sharp1^{+/+}$ and $Sharp^{-/-}$ mice. The frequency of mature granulocytes ($Gr1^+$ $CD11b^+$) and B cells ($B220^+$) was also unchanged (Fig. 5c and Supplementary Fig. 5b).

To investigate the reconstitution ability of $Sharp1^{-/-}$ hematopoietic stem and progenitor cells (HSPCs), we performed competitive transplantation assays by injecting 500,000 BM cells from $Sharp1^{+/+}$ or $Sharp1^{-/-}$ mice into lethally irradiated (900 rads) $CD45.1^+CD45.2^+$ congenic mice along with equal number of BM cells from $CD45.1^+$ congenic mice. In PB chimerism analysis, no differences were observed in percentage of donor cells in myeloid, B, and T cell lineages between recipients of $Sharp1^{+/+}$ and $Sharp1^{-/-}$ BM cells over a period of 16 weeks after the transplantation (Fig. 5d, Supplementary Figs. 5c and 5d). Collectively, these results demonstrate that $Sharp1$ deletion does not affect steady-state hematopoiesis, as well as the ability of HSPCs to differentiate into multi-lineage cells and reconstitute hematopoiesis.

**SHARP1 binds to actively transcribed genes.** Given these findings and the known functions of SHARP1 as a bHLH transcription factor, we hypothesized that SHARP1 binds to target genes and regulates their expression, which are important for the development and maintenance of AML. To delineate the direct transcriptional target genes, we performed ChIP-seq using antibodies against SHARP1 in ML-2 cells and identified 7,443 SHARP1-bound genes. Consistent with the known binding to E-box with high affinity as a homodimer[25,26], CACGTG was the most enriched motif in the binding regions across the genome, and it was increased near the binding peaks (Fig. 6a and Supplementary Fig. 6a). A large proportion of SHARP1 occupancy was located at the proximal promoter ($-1$ kb, $+100$ bp from TSS, 36%), intronic (28%), and intergenic regions (27%) (Fig. 6a). SHARP1 was considered to function as a transcriptional repressor, either by direct or indirect binding to DNA, and interacts with DNA-bound transcription factors, such as C/EBP or MyoD, and recruits G9a, HDAC1, and SIRT1 to binding sites, resulting

in alteration of histone modifications[27–29]. To delineate chromatin accessibility within SHARP1 binding sites, we overlaid them with the regions enriched with active enhancer and promoter marks (H3K4me3 and H3K27ac)[30] and the repressive mark (H3K27me3)[31]. Remarkably, SHARP1 binding sites were enriched in H3K4me3 and H3K27ac marks within the gene loci (Fig. 6a), defined as the promoter region ($-2$ kb) and gene body, of highly expressed transcripts (H3K4me3; 6459 genes, H3K27ac; 5840 genes), whereas the binding sites enriched in the H3K27me3 sites are located within the gene loci of poorly transcribed genes (1055 genes) (Fig. 6b). The genes that were known to be bound by SHARP1 protein (CLOCK, PER1, SHARP2, and MLH1)[32,33] demonstrated a SHARP1 peak in their promoters (Fig. 6c).

**SHARP1 regulates target genes in MLL-AF6 AML cells.** As the majority of H3K27ac marks overlaps H3K4me3 profiles, we focused our attention on gene loci enriched in H3K4me3, a known active enhancer and promoter mark[34]. Interestingly, biological pathway analysis revealed a significant enrichment in genes related to metabolic pathways (adj. $p = 1.71E-89$), cell cycle (adj. $p = 1.68E-19$), ribosome biogenesis (adj. $p = 8.74E-9$), and DNA replication (adj. $p = 8.76E-7$) (Fig. 6d), indicating that SHARP1 is involved in diverse biological processes crucial for AML cells. Moreover, we performed RNA-seq analysis comparing expression profiles of ML-2 control to SHARP1 knockdown cells and found that 319 genes of SHARP1-bound genes were downregulated and 326 genes were upregulated upon SHARP1 knockdown (Supplementary Fig. 6b). The downregulated genes were associated with cell cycle, TGF-β signaling, FoxO signaling, HIF-1 signaling, and cancer (CDKN1B, FLT3, PDK1, FOXO1, BCL2, ERG) (Supplementary Fig. 6c), suggesting potential positive regulation of these pathways by SHARP1 to maintain MLL-AF6 AML activity.

**SHARP1 does not influence circadian clock genes expression.** SHARP1 is one of the regulators of the mammalian molecular clock[33]. Circadian clock genes generally play a critical role in cancer cells with tumor suppressive or oncogenic properties in a context-dependent manner[35,36]. In leukemias, PER2 (period circadian clock 2) was identified as a downstream target of C/EBPα and had its genitive impact in promoting AML initiation[37]. A recent study demonstrated that perturbation of the core circadian protein heterodimer, CLOCK/BMAL1, induced myeloid differentiation of AML cells and depleted LSC, highlighting the importance of clock genes in AML[38]. SHARP1 is regulated by the CLOCK/BMAL1 and represses their transcriptional activity by competing for DNA binding or direct interaction with BMAL1[33]. Thus, SHARP1 functions as a negative regulator for PER1, SHARP2, and SHARP1 itself in a feedback loop. Consistently, SHARP1 was bound to the promoter of the circadian clock genes (CLOCK, PER1, and SHARP2) in ML-2 cells (Fig. 6c, d). We asked whether upregulated SHARP1 induce the aberrant expression of the clock genes, which have a potential to affect AML activity. We investigated the expression of ten circadian clock genes (SHARP2, BMAL1, CLOCK, CRY1, CSNK1E, PER1, PER2, PER3, CUL1, and NR1D) in AML patients, comparing MLL-AF6 to other MLLr or non-MLLr AML, none of which exhibited aberrant expression (Supplementary Fig. 6d). This suggests that SHARP1 does not affect the expression of other clock genes in MLL-AF6 AML cells despite their interlocked feedback control in other physiological contexts.

**SHARP1 cooperates with MLL-AF6 to regulate target genes.** Having established that SHARP1 could contribute to

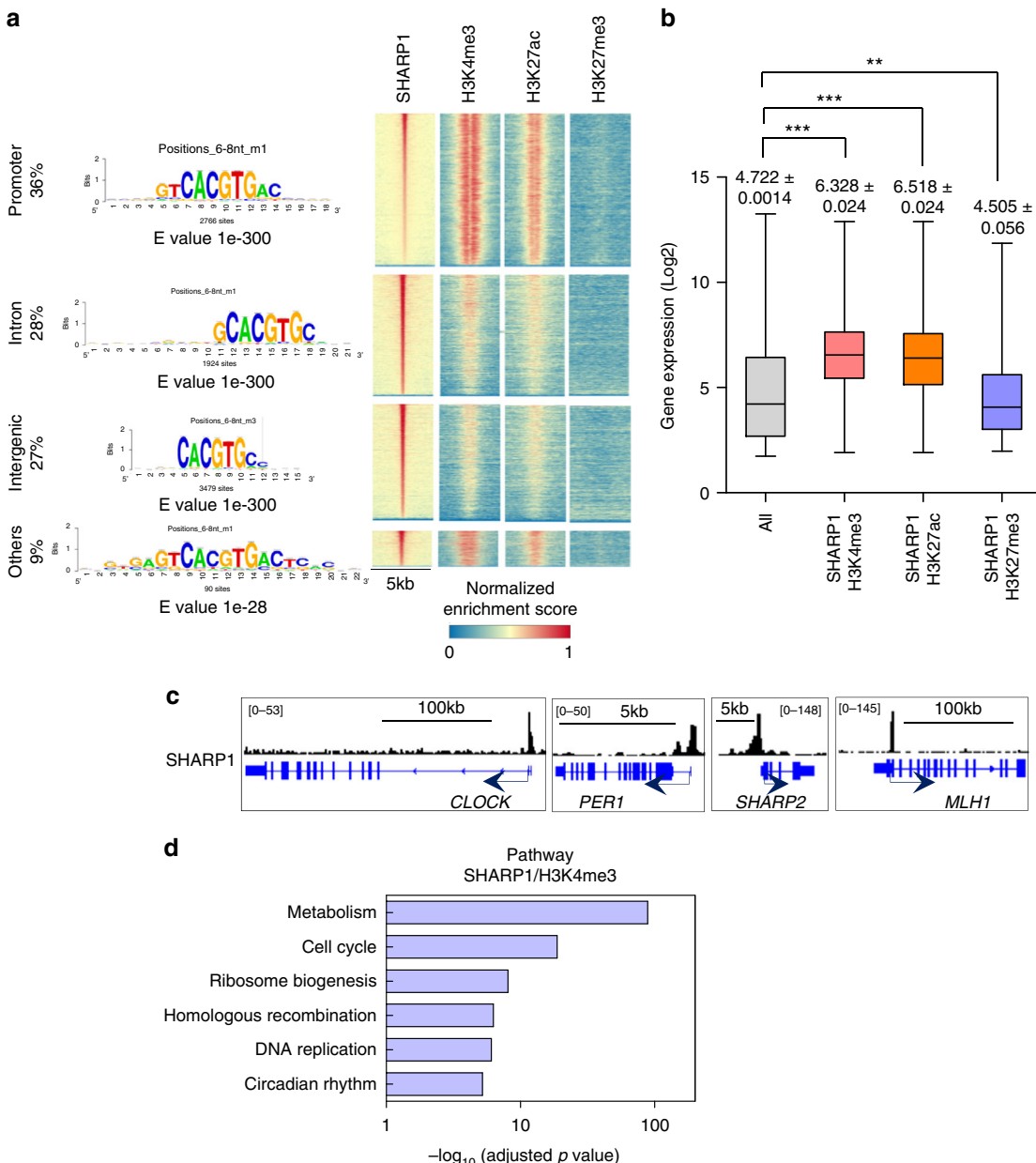

**Fig. 6** SHARP1 binds to actively transcribed genes and positively regulates target genes. **a** Integrated view of SHARP1 binding sites in conjunction with H3K4me3, H3K27ac, and H3K27me3 profiles across promoters, introns and intergenic regions. Top enriched motifs within SHARP1 ChIP-seq peaks in ML-2 cells are shown according to their genomic location discovered by peak-motifs module from the RSAT suite, using oligomer length ranging from 6 to 8 nucleotides and the 'merge lengths for assembly' option. **b** Box plot showing the expression levels in microarray analysis of ML-2 cells for the all genes (25293 genes) and genes enriched with SHARP1 + H3K4me3 (6459 genes), SHARP1 + H3K27ac (5840 genes), and SHARP1 + H3K27me3 (1055 genes) identified in ChIP-seq analysis. The box extends from the 25th to 75th percentiles and the whisker extends from the minimum level to the maximum. Median value is plotted in the box. **c** Representative SHARP1 binding peaks in the known target gene loci (circadian clock genes and *MLH1*). **d** Pathway analysis for the genes in the SHARP1 and H3K4me3 co-bounded regions within the promoter and gene body

development and maintenance of MLL-AF6 AML, we next sought to determine whether it cooperates with MLL-AF6 to regulate transcription of target genes. Intriguingly, Gene Set Enrichment Analysis (GSEA) revealed that the downstream genes of HOXA9/MEIS1, MLL, and MLL-AF4 are enriched in those downregulated upon SHARP1 knockdown (Fig. 7a). To delineate the correlation of genome-wide occupancy between SHARP1 and MLL-AF6, we overlaid SHARP1 and MLL$^N$ bound regions obtained from ChIP-seq analysis. Notably, 78 out of 92 MLL-AF6 target genes were SHARP1-bound (Fig. 7b, Supplementary Table 3), and 14 of the co-target genes were downregulated upon

SHARP1 knockdown (Fig. 7c), whereas none of these genes were upregulated (≥1.5-fold). The gene set includes previously identified oncogenic targets of MLL-FPs, such as *MEF2C*[15], *CDK6*[39], and *RUNX2*[8], which demonstrated co-localization of MLL, MEN1, and SHARP1 within the promoter loci, accompanied with enrichment of H3K79me2/3 (Fig. 7d). These results suggest that SHARP1 cooperates with the MLL-AF6 protein complex, and expression of some MLL-AF6 target genes depend on SHARP1.

To determine whether SHARP1 forms a complex with MLL-AF6, we carried co-immunoprecipitation (co-IP) experiments in nuclear extracts from MLL-AF6 cell lines ML-2 and SHI-1. AF6

or MLL[N] co-IPs failed to detect SHARP1, which may have been obscured by heavy chain bands. To circumvent this issue, we performed co-IP in 293 T cells that were transfected with MLL-AF6 and SHARP1, and demonstrated a robust interaction between the two proteins (Fig. 7e). Intriguingly, we also observed interaction between SHARP1 and MLL-AF9 (Supplementary Fig. 7a), indicating that SHARP1 interacts with the portion of MLL that is present in both MLL fusions. Using a series of MLL deletion mutants[14], we identified a region, amino acids 541–1251 (541–1251aa) of MLL, which was responsible for interaction with SHARP1. We did not observe an interaction with MLL (1–540aa),

while the interactions with other MLL mutants were comparable to that of MLL (1–1251aa) (Fig. 7f, g), indicating that SHARP1 interaction with MLL-AF6 and MLL-AF9 is dependent on MLL (541–1251aa). This region contains the transcriptional repression domains, including a DNA methyltransferase domain (MT) that shares homology to methyl DNA-binding proteins[40,41] and recruits repressor complexes containing HDAC1[41]. Given these findings, it is conceivable that the interaction with SHARP1 could alter the constituents of the MLL-AF6 complex and influence the regulation of target genes. Although SHARP1 interacts with common portion of MLL-FP, its specific expression in MLL-AF6

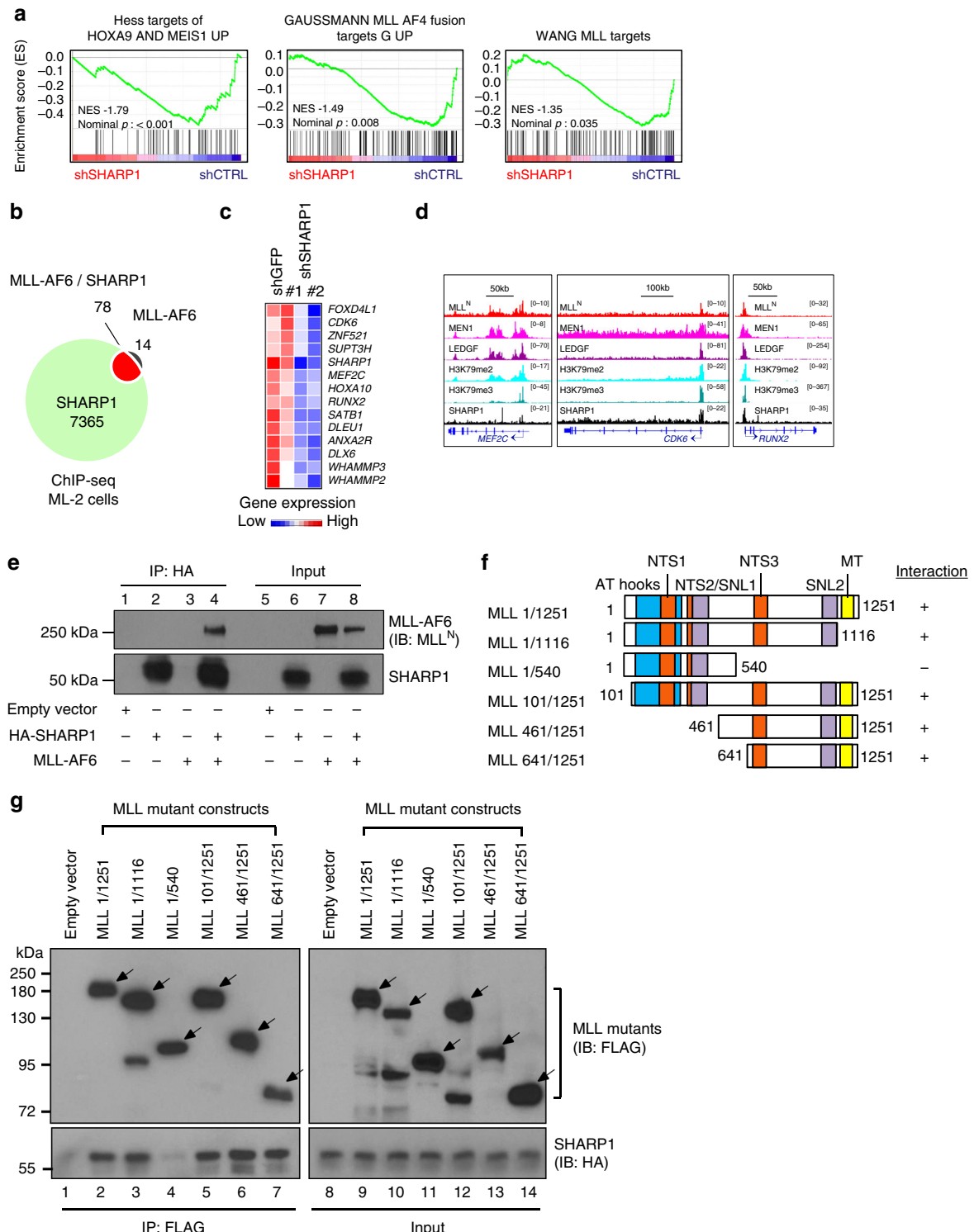

only might provide a unique mechanism in regulation of the MLL-AF6 target genes.

## Discussion

Our study reveals a unique mechanism in the leukemogenicity in MLL-AF6 AML. We identified direct MLL-AF6 target genes that are overexpressed in MLL-AF6 AML patients compared to other subtypes of MLLr-AMLs and focused on the bHLH transcription factor, SHARP1, the highest and most significantly upregulated gene. We demonstrated that SHARP1 plays an oncogenic role to maintain clonogenic growth and leukemia-initiating potential, regulating the expression of genes crucial for leukemia cell survival including MLL-AF6 target genes (Fig. 8).

MLLr-leukemias present with related gene expression profiles as a result of a common MLL-FPs driven activation of transcriptional elongation machinery[20,42]. Recent genome-wide ChIP-seq analyses identified various sets of direct target genes of MLL-FP, which consistently included the master regulatory factors (HOXA7, HOXA9, HOXA10, and MEIS1) required for the development of MLLr-leukemias. In this study, we identified 101 genes bound by MLL-AF6, and this number is comparable to the previous studies of other MLL-FPs, in which 165 genes were identified in MLL-AF4[43], 139 genes in MLL-AF9[8], and 178 genes in MLL-ENL[44]. In contrast to the recent ChIP-seq analysis which used ML-2 cells[45], using a different MLL[N] antibody, we were able to identify SHARP1 as a direct target of MLL-AF6, and validate this finding by ChIP-qPCR. Furthermore, we found that MEN1 and LEDGF, two major MLL-FP subunits, co-bound to SHARP1 gene loci, corroborating our findings.

MLL-FPs recognize their target gene loci through the CXXC domain of MLL and the PWWP domain of LEDGF[14,21,45,46]. The CXXC domain specifically binds to unmethylated CpG DNA, which is enriched in active promoters, whereas the PWWP domain recognizes H3K36me2/3, which generally associates with transcriptionally active regions. Although MLL-FPs functions as an epigenetic reader through these common subunits of the MLL-FP complex, the majority of MLL-AF6 targets are not included in the gene sets of MLL-ENL, MLL-AF9, or MLL-AF4 targets[8,43,44], raising the possibility that unknown mechanisms may be involved in this process. It is conceivable that distinct cellular functions and localizations of translocation partners may determine the unique target genes. AF6 (MLLT4), also known as afadin, is the most common MLL cytoplasmic partner protein and has a dual residency protein in the plasma membrane and the nucleus[47]. AF6 may be involved in the transcription of unique MLL-AF6 target genes by recruiting transcription factors or co-activators within the nucleus[48].

Given the high expression level of SHARP1 comparable to those of pivotal oncogenic target genes, HOXA9 and MEIS1, in MLL-AF6 AML patients, we hypothesized that SHARP1 plays an oncogenic role in MLL-AF6 AML cells. SHARP1 may exert contextual tumor suppressive or oncogenic functions, depending on the type of cancer. A recent study demonstrated that SHARP1 is highly expressed in renal cell carcinoma cells and its over-expression accelerated tumor progression in xenograft models[49], whereas in triple negative breast cancers, SHARP1 mediates the anti-metastatic function of p63 by degrading HIF-1α, and its overexpression is associated with a favorable prognosis[50]. In physiological conditions, SHARP1 is expressed in various tissues, though the expression level is generally low and is upregulated by external stimuli such as cytokines, infection, and hypoxia[26], indicating its potent role as a positive regulator for cell survival under stress conditions. A recent study demonstrated that SHARP1 expression can be induced by DNA-damaging agents, and that SHARP1 inhibits activation of the p53 pathway including pro-apoptotic genes[51], providing protection from cytotoxic effects. In agreement with the anti-apoptotic role of SHARP1, SHARP1 knockdown resulted in robust apoptosis in human MLL-AF6 AML cells, accompanied by the upregulation of p53 pathway and apoptosis associated genes (Supplementary Fig. 6e). However, a recent study by Coenen et al. demonstrated that shRNA-mediated SHARP1 knockdown did not have any effect in SHI-1 cells[52], in contrast to our findings. This discrepancy might be explained by the difference in knockdown efficiencies with the use of different shRNAs against SHARP1. In fact, one of the shRNAs was common between their study and ours, and has led to reduced growth and increased apoptosis in SHI-1 cells, even though the differences were only significant in our study. Also, cell lines may acquire mutations that alter original characteristics after long periods of culture, which could explain differences in knockdown between these two studies. Based on our findings in the three MLL-AF6 and the two other MLLr-AML cell lines, as well as genetic deletion in murine AML models, we concluded that SHARP1 plays an oncogenic role in MLL-AF6 AML cells.

In contrast to a number of evidence for a transcriptional repressive role[25,27,28,53,54], SHARP1 activates JunB and Gata3 expression to induce naïve T cells to Th2 T cells, and genetic deletion of Sharp1 leads to reduced histone H3 acetylation at the JunB conserved non-coding sequence and Gata3 promoter[55]. This indicates that Sharp1 regulates chromatin modification at these two loci and functions as a transcriptional activator. We demonstrated that SHARP1 binds to E-box motifs in active chromatin that are marked by H3K4me3 and H3K27ac, which suggests an interaction between SHARP1 and transcriptional

**Fig. 7** SHARP1 interacts with MLL-AF6 and regulates gene targets. **a** Enriched gene sets in ML-2 shGFP cells over shSHARP1 on RNA-seq. **b** Venn diagram showing overlapping of SHARP1-bound genes with MLL-AF6 target genes (MLL[N]+H3K79me2) in ML-2 cells. **c** Heatmap images representing the relative expression levels of 14 MLL-AF6/SHARP1 co-target genes downregulated upon SHARP1 knockdown obtained from RNA-seq data. **d** Genome view of MLL[N], MEN1, LEDGF, H3K79me2, H3K79me3, and SHARP1 peak binding on three MLL-AF6 + SHARP1 target genes in ML-2 shGFP: MEF2C, CDK6 and RUNX2 gene. **e** Co-immunoprecipitation studies of SHARP1 and MLL-AF6 with an anti-HA antibody in 293 T cells transfected with plasmids encoding MLL-AF6 and/or HA-tagged SHARP1. Proteins present in immunoprecipitates (IP, lane 1–4) or whole cell lysates of transfected cells (input, lane 5–8) were separated by SDS-PAGE and immunoblotted with antibodies specific for MLL[N] and SHARP1. Interaction of SHARP1 and MLL-AF6 was detected (lane 4) and not observed in negative control lanes with either empty vector, SHARP1 or MLL-AF6 only (lane 1–3). **f** Schematic showing a series of MLL deletion mutants. Interaction with SHARP1 is indicated by + sign and loss of interaction by − sign. Boxes indicate AT hook motifs (blue), nuclear translocation sequences (NTS1 and NTS2) (orange), subnuclear localization domains (SNL1 and SNL2) (purple), and DNA methyltransferase domain (MT) (yellow). **g** Domain mapping analysis of MLL required for interaction with SHARP1. 293 T cells are transfected with plasmids encoding FLAG-tagged MLL deletion mutants and HA-tagged SHARP1. Whole cell lysates were prepared from the transfected cells and subjected to immunoprecipitation with anti-FLAG antibody. Proteins present in immunoprecipitates (IP, lane 1–7) or whole cell lysates of transfected cells (input, lane 8–14) were separated by SDS-PAGE and immunoblotted with antibodies specific for FLAG-tagged MLL deletion mutants and HA-tagged SHARP1. The arrows indicate MLL mutant proteins. Western blots are representative of at least three independent experiments

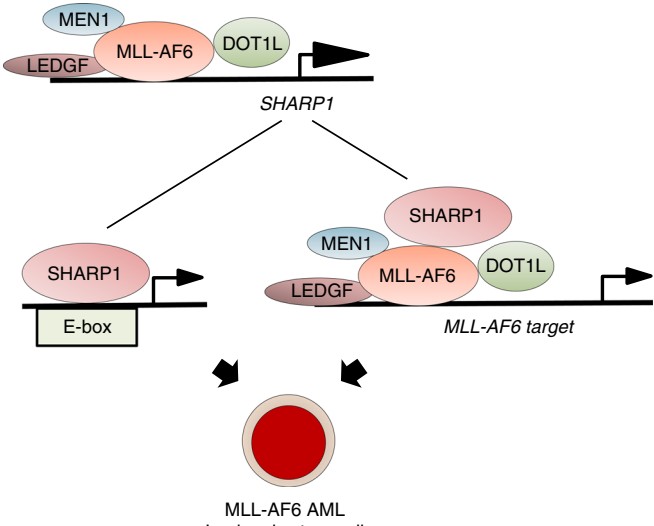

**Fig. 8** Oncogenic role for SHARP1 in MLL-AF6 AML MLL-AF6 protein binds to and activates the *SHARP1* gene in a DOT1L-dependent manner. Upregulated SHARP1 binds to E-box motifs in active chromatin, and also interacts with MLL-AF6 to regulate a subset of genes critical for leukemogenicity. This unique transcriptional machinery contributes to the maintenance of MLL-AF6 AML leukemic stem cells

activators or co-activators, recruiting chromatin modifiers to the binding sites. This process can be influenced by differences in post-translational modifications of SHARP1 protein, which allows association with distinct protein complexes in different cell contexts. For instance, SHARP1 sumoylation regulates its association with G9a[56], which determines the SHARP1-dependent functions. We further demonstrated that physical interaction of SHARP1 and MLL-AF6 and co-localization on the promoters of the majority of the MLL-AF6 target genes, some of whose expressions are sensitive to SHARP1 levels. It is also possible that SHARP1 regulates these genes through interaction with constituents of these complexes. Delineating the proteins interacting with SHARP1 in MLL-AF6 AML cells will provide further insights into the process of gene regulation by SHARP1 in leukemia.

We identified a subset of (a) SHARP1 targets and (b) co-targets of MLL-AF6 and SHARP-1 that are critical for leukemogenicity using an integrative analysis of RNA-seq and ChIP-seq datasets in ML-2 cells. However, these genes are not overexpressed in MLL-AF6 AML patients compared to the other subtypes of MLLr-AML. SHARP1 ChIP-seq analysis highlighted that various motifs of potential co-factors are enriched in the promoter of SHARP1 targets (Supplementary Fig. 7B), suggesting a more complex regulatory mechanism involving other transcription factors. It is plausible that those genes are activated by other transcription factors in different AML subtypes that generally do not express SHARP1. It will be of future interest to investigate how overexpressed SHARP1 influences the recruitment of transcriptional regulatory factors to chromatin, providing a unique mechanism for gene regulation in ML-AF6 AML.

Notably, SHARP1 played more profound role in maintaining the leukemia-initiating potential of L-GMP than whole BM cells in MLL-AF6 AML. This suggests that the pathways or genes that SHARP1 regulates may differ with the differentiation stage of leukemia cells, and may play a more significant role in LSC activity. MLL-AF6 AML patients present with a dismal clinical prognosis due to resistance to initial chemotherapy and high rate of relapse[57], which may be caused by residual chemotherapy-

resistant quiescent LSC[58]. Our finding that *Sharp1* is dispensable for normal HSPC function suggests that SHARP1 could be a promising therapeutic target of MLL-AF6 AML LSC. Additionally, we showed that SHARP1 expression is sensitive to treatment with DOT1L inhibitor, indicating that inhibition of DOT1L could be a promising therapeutic approach to eliminate LSC and improve the prognosis in these patients by preventing relapse after chemotherapy.

## Methods

**Microarray Data.** Gene expression data of AML patients were obtained from GSE19577, GSE14468, and GSE61804, and NBM CD34+ cells were from GSE19429, all from the NCBI Gene Expression Omnibus (GEO) database. The probe-set expression data were generated using robust multichip average (RMA) and then normalized using the cross-correlation method[59]. Differentially expressed genes were then derived using the $\log_2$ fold change cutoff of 0.5 and the (t-statistic) $p$ value cutoff of 0.05. For each gene, the $p$ value between 14 cases of MLL-AF6 and 42 cases of other subtypes of MLL-rearranged AMLs [MLL-AF9 ($n = 16$), MLL-AF10 ($n = 12$), MLL-ENL ($n = 4$), MLL-ELL ($n = 3$), MLL-SEPTIN6 ($n = 3$), MLL-AF4 ($n = 2$), and MLL-AF1q ($n = 2$)] samples against the mean of $\log_2$ fold change was used to generate volcano plots. Unsupervised hierarchical gene-expression clustering of AML cells from GSE1159 and GSE6891 was performed as described previously[20].

**Cell lines and DOT1L inhibitor.** HEK293T (ATCC, CCL11268) and BOSC23 (ATCC, CRL11270) cells were maintained in DMEM (Biowest) supplemented with 10% heat-inactivated FBS (Biowest). ML-2 (DMZ, ACC 15), CTS[60], SHI-1 (DMZ, ACC-645), MOLM-14 (DMZ, ACC 777), MV4-11 (ATCC, CRL9591) and Kasumi-1 (ATCC, CRL2724) cells were maintained in RPMI-1640 (Biowest) supplemented with 10% heat-inactivated FBS (Biowest). DOT1L inhibitor EPZ5676 (Epizyme) was prepared in DMSO at 10 mM stock solution. ML-2, SHI-1, MOLM-14, MV4-11, and Kasumi-1 identities were confirmed by STR profiling (Genetica DNA Laboratories, NC, USA). All cell lines were tested negative for mycoplasma contamination by MycoAlert PLUS mycoplasma detection kit (Lonza).

**shRNA lentivirus transduction and transplantation.** Lentiviral plasmids (pLKO.1-puro) encoding shRNA targeting SHARP1 or MLL[N] were either obtained from Sigma MISSION [TRCN0000016946 (shSHARP1 #2), TRCN0000005954 (shMLL #2), TRCN0000234741 (shMLL#1)], or cloned into pLKO.1 (sh sequence: CGAGACGACACCAAGGATA, shSHARP1 #1)[50]. Lentivirus packaging was performed in 293 T cells by co-transfecting shRNA lentiviral plasmids with pCMV-dR8.91and pCMV-VSVG using Lipofectamine 2000 (Invitrogen). ML-2, CTS, SHI-1, MOLM-14, and MV4-11 cells were exposed to viral particles with multiplicities of infection (MOI) ranging from 1 to 2, in the presence of 6 μg/mL polybrene (Santa Cruz) for 24 h. Cells were selected in media containing 0.5 μg/mL puromycin at 48 h post-transduction, and checked for knockdown efficiency by qPCR and immunoblotting at 5 to 7 days post-transduction and then $5 \times 10^4$ ML-2 and $1 \times 10^5$ CTS cells were injected into sublethally irradiated NSG mice.

**Cell growth and Apoptosis assay.** ML-2, CTS cells, and SHI-1 cells were transduced with lentiviral shRNA, shGFP, shSHARP1#1, or shSHARP1#2, selected with puromycin (0.5 μg/mL), and $4 \times 10^4$ cells seeded per well in 96-well plates and counted every 2 days by hemocytometer. Trypan blue was used to exclude dead cells. Apoptosis assay were performed according to the manufacturer's instructions using the FITC Apoptosis Assay Kit (BD Biosciences) and analyzed by LSRII flow cytometer.

**Serial replating assay and CFC assay.** Serial replating assays were performed by plating 20,000 cells of murine LSK cells transduced with MLL-AF6 on methylcellulose M3234 (Stem Cell Technologies) supplemented with 6 ng/mL interleukin (IL)-3, 10 ng/mL IL-6, and 20 ng/mL SCF (Peprotech). Colony numbers were counted every 7 days and subjected to replating. Colony-forming cell (CFC) assays were performed by plating 5,000 ML-2 or CTS cells transduced with the indicated shRNAs on methylcellulose H4531 (Stem Cell Technologies) after puromycin selection (0.5 μg/mL). Colony numbers were counted after 7 days.

**Mice.** Mice were housed at in a sterile barrier facility within the Comparative Medicine facility at the National University of Singapore. All mice experiments performed in this study were approved by Institutional Animal Care and Use Committee (IACUC). *Sharp1*[−/−] mice were described previously[23]. CD45.1+ congenic mice (B6.SJL) and NSG mice were purchased from Jackson Labs.

**Retrovirus transduction and generation of leukemia.** The MLL-AF6 construct, consisting of 35-347 amino acids of the AF6 portion[12], was cloned into MSCV-puro. MLL-AF6 and MLL-AF9[15] retroviruses were produced in BOSC23 cells. For transduction, FACS-sorted LSK cells were seeded in Retronectin (Takara)-coated

dishes containing retroviral supernatants for 48 h and the transduced cells were expanded for 7 days in methylcellulose M3234 (Stem Cell Technologies) supplemented with cytokines (6 ng/mL IL-3, 10 ng/mL IL-6, and 20 ng/mL SCF). For MLL-AF6 AML, after puromycin selection (2 µg/mL) $2 \times 10^5$ cells were injected into sublethally (650 rads) irradiated CD45.1 congenic mice to generate leukemia. For MLL-AF9 AML, $2 \times 10^5$ cells after transduction were injected into sublethally irradiated congenic mice. For generation of secondary leukemia, $2 \times 10^5$ primary leukemic cells were injected into sublethally (650 rads) irradiated congenic mice. PB was obtained every month after transplantation and analyzed by Hemavet HV950FS (Drew Scientific) and FACS. Cells were stained with anti-CD45.1 and CD45.2 antibodies to distinguish donor-derived cells from the host cells, as well as anti-CD11b and Gr1 antibodies to identify leukemia cells. A recipient mouse was considered positive if donor-derived cells were present and also constituted more than 0.3% of the cells in the PB. BM cells harvested from moribund mice were cytospun and stained with Giemsa's azur-eosin-methylene blue (Merck Millipore).

**Limiting dilution assay.** Unfractionated BM (2000, 200, 100) or FACS-sorted L-GMP (500, 50, 5) cells from leukemic mice were injected into sublethally (650 rads) irradiated congenic mice. PB was obtained each month after transplantation and analyzed by FACS for chimerism. Calculation of the frequency of LSC and the statistical $p$ value was performed using the extreme limiting dilution analysis (ELDA) online software (http://bioinf.wehi.edu.au/software/elda/). A recipient mouse was considered positive if CD45.2$^+$ cells constituted more than 0.3% of the all nucleated cells in the PB.

**Competitive transplantation assay.** 500,000 unfractionated BM cells from CD45.2$^+$ Sharp1$^{+/+}$ or Sharp1$^{-/-}$ mice were injected into lethally (900 rads) irradiated CD45.1$^+$ and CD45.2$^+$ congenic mice along with an equal number of CD45.1$^+$ BM cells. PB was obtained every month after transplantation and analyzed by FACS. Cells were stained with anti-CD45.1 and CD45.2 antibodies to distinguish donor-derived cells from the host cells, as well as lineage-specific antibodies CD11b, Gr1, B220, CD4 and CD8 to identify myeloid, B, and T lineages.

**Quantitative PCR.** RNA was extracted using RNeasy kit (QIAGEN), reverse transcribed using the QuantiTech Reverse Transcription kit (QIAGEN), and quantitatively assessed using an ABI7500 (Applied Biosystem). For each sample, transcript levels of tested genes were normalized to β-actin using the delta CT method. The highest expression was arbitrarily set to 1 and expressions in the other samples were normalized to this value. All experiments were performed with three replicates. PCR was performed on cDNA using following primers: MLL-AF6 fusion gene: 5′-GTCCAGAGCAGAGCAAACCAG-3′, 5′-CTGACATGCACTTCATAGAGTG-3′, SHARP1 (human): 5′-TAACCGAGCAACAGCATCAG-3′, 5′-TTTGAAATCCCGAGTGGAAC-3′, HOXA9 (human): 5′-GTATAGGGGCACCGCTTTTT-3′, 5′-AATGCTGAGAATGAGAGCGG-3′, β-ACTIN (human): 5′-ACCCTGAAGTACCCCATCGA-3′, 5′-CTCAAACATGATCTGGG-3′, Sharp1 (mouse): 5′-ACCGAATTAATGAATGCATTGCTCAG-3′, 5′-GTAAATACACCCCGGAGTCCATCA-3′, β-actin (mouse): 5′-GGTCCACACCCGCCACCAG-3′, 5′-TTGCTCTGGGCCTCGTCACC-3′.

**Chromatin immunoprecipitation assays.** Chromatin immunoprecipitation (ChIP) (for both qPCR and sequencing) were performed as previously described[61], with the following modifications: ML-2 cells were fixed using a 1% formaldehyde (FA) fixation protocol for 10 min for histone marks and SHARP1. For the other proteins, the cells were fixed using a 1% FA fixation protocol for 10 min, while a 45 min, 2 mM disuccinimidyl glutarate (DSG) and a 30 min 1% FA double fixation protocol was used. The antibodies used included SHARP1 (a mixture of H-72 Santa-Cruz, 12688-1-AP Proteintech, and ab175544 Abcam), H3K79me3 (Diagenode, pAb-068-050), MLL1 (Bethyl, A300-086A), H3K4me3 (Active Motif, 39159), H3K27me3 (Millipore, 07-499), H3K27ac (Millipore, 07-360), LEDGF (Bethyl, A300-848A), and MEN1 (Bethyl, A300-105A). Fixed chromatin samples were fragmented using a Bioruptor sonicator (Diagenode) for 30 min at high power in a constantly circulating 4 °C water bath to an average size of 200-500 bps. Antibody:chromatin complexes were collected with a mixture of protein A and Protein G Dynabeads (Life Technologies) collected with a magnet, and washed 2 × with a solution of 50 mM HEPES-KOH, pH 7.6, 500 mM LiCl, 1 mM EDTA, 1% NP-40, and 0.7% Na-Deoxycholate. After a TE wash, samples were eluted, RNase and Proteinase K treated, and purified using a QIAGEN PCR purification kit. The primer sequences were HOXA9 promoter: 5′-TGGCTGCTTTTTTATGGCTTCA ATTATTG-3′/5′-CCGCGTGCGAGTGCG-3′, GAPDH promoter: 5′-CCCCTCC TAGGCCTTTGC-3′/5′-GCTGAGAGGCGGGAAAGTT-3′, and SHARP1 promoter: 5′-TGGGTAAACTTGAGTCCCAAAGGAAATT-3′/5′-TGCAAGTTG CTTCTTCTCGGAGGC-3′. ChIP samples were quantified relative to inputs as described[61]. Briefly, the amount of genomic DNA co-precipitated with antibody is calculated as a percentage of total input using the following formula $\Delta CT = CT$ (input) − CT (chromatin IP), % total $= 2^{\Delta CT} \times 5.0\%$. 50 µl aliquot taken from each of 1 ml of sonicated, diluted chromatin before antibody incubation serves as the input, thus the signal from the input samples represents 5% of the total chromatin used in each ChIP. CT values were determined by choosing threshold values in the linear range of each PCR reaction. The % input values of MLL$^N$ and H3K79me2

enrichment on HOXA9 (positive control) and SHARP1 loci were normalized against that of the GAPDH locus (negative control) in the respective samples for comparison.

**RNA/ChIP-sequencing.** RNA-seq libraries were prepared using Illumina Tru-Seq Stranded Total RNA with Ribo-Zero Gold kit protocol, according to the manufacturer's instructions (Illumina, San Diego, California, USA). Libraries were validated with an Agilent Bioanalyzer (Agilent Technologies, Palo Alto, CA), diluted, and applied to an Illumina flow cell using the Illumina Cluster Station. ChIP-seq libraries were prepared using Next ChIP-Seq library prep reagent set (New England Biolabs), and multiplexed (New England Biolabs). Each library was sequenced on an Illumina Hiseq2000 sequencer.

**ChIP-Seq data analysis.** In-house ChIP-seq data for SHARP1, MEN1, LEDGF, H3K79me3, H3K4me3, H3K27me3, and H3K27ac in ML-2 and MLL$^N$ in SHI-1 were used in this study. ChIP-seq data for MLL$^N$ and H3K79me2 in ML-2 were obtained from GSE83671, and those in MV4-11 and THP-1 were from GSE79899. Sequencing quality check for every dataset was performed. Mapping of the reads was performed using BWA to human genome version UCSC Hg19. MACS2 was used for peak calling between the special ChIP samples and the corresponding inputs with filtering parameters of cutoff $p$ value at 0.05, minimal peak fold enrichment of 5, and minimal peak height of 10 reads. Peak integration cross different ChIP-seq datasets was done by utilizing the Bedtools. The Database for Annotation, Visualization and Integrated Discovery (DAVID) v6.7 was used to detect potential significantly altered pathways (https://david.ncifcrf.gov/). SHARP1 DNA binding motif enrichment was computed using the peak-motifs module from the RSAT suite[62] using oligomer length ranging from 6 to 8 nucleotides and the "merge lengths for assembly" option. ChIP-seq heatmaps were generated using ChAsE vers. 1.0.11[63].

**RNA-Seq data analysis.** RNA-seq was performed for duplicates of shGFP, shSHARP1#1 and shSHARP1#2. After mapping of RNA-seq data to human genome version UCSC Hg19, normalization among the samples was performed using the total mappable counts. For identification of enriched gene sets or pathways, we utilized the GSEA (Gene Set Enrichment Analysis) software with the tool for preranked gene list (http://www.broad.mit.edu/gsea/) and the database MSigDB. v5.0.

**Co-immunoprecipitation.** HEK293T cells were transfected with the following plasmids: empty vector (control), HA-tagged SHARP1 (human), MLL-AF6, MLL-AF9[15], and/or MLL deletion constructs[14]. The cells were lysed in co-immunoprecipitation (co-IP) lysis buffer (10 mM Tris-Cl pH 7.5, 150 mM NaCl, 0.5 mM EDTA and 0.5% NP-40), supplemented with complete EDTA-free protease inhibitor cocktail (Roche) for 30 min on ice. Lysates were cleared at 15,000× $g$ for 15 min at 4 °C. Co-immunoprecipitation was performed in co-IP lysis buffer with 2 µg of anti-HA (Santa Cruz, sc-7392) or anti-FLAG antibodies (Sigma, F1804) for 3 h or overnight at 4 °C. Following which, three washes of immunoprecipitated proteins were performed with wash buffer (10 mM Tris-Cl pH 7.5, 150 mM NaCl, and 0.5 mM EDTA). The bound proteins were eluted with 2 ×laemmli buffer for 5 min at room temperature with constant shaking. Samples were resolved on 7% SDS-PAGE and subjected to Western blot analysis.

**Immunoblotting, immunoprecipitation, and antibodies.** For whole cell lysis, 1 × 10$^7$ cells were lysed with radioimmunoprecipitation (RIPA) buffer. Protein concentrations were quantitated with Biorad Protein Assay (Bio-Rad). Proteins were separated on 10% SDS-PAGE gels. Immunoblots were incubated with primary antibody overnight at 4 °C, followed by a secondary horseradish peroxidase (HRP)-conjugated antibody (1:2000 dilution) at room temperature for 1 h. Details on the antibodies used and dilutions are described in Supplementary Table 5. Full blots are shown in Supplementary Fig. 8.

**Flow cytometry.** Single-cell suspensions were analyzed by flow cytometry. Antibodies conjugated with phycoerythrin (PE), PE-CY7, fluorescein isothiocyanate (FITC), allophycocyanin (APC), APC-CY7, and Pacific Blue obtained from BD Pharmingen, BioLegend, and eBioscience were used. Details on the antibodies used and dilutions are described in Supplementary Table 4. Stained cells were analyzed with an LSRII flow cytometer and sorted using a FACS Aria (BD Biosciences). Flow Jo 7.5 (Tree Star) was used for data acquisition and analysis.

**Statistical information.** The statistical significances were assessed by two-sided Student's unpaired $t$-test using the GraphPad PRISM 5 unless otherwise specified.

**Data availability.** The data supporting the findings of this study are available in the article or supplementary information files. Any other relevant data are available from the authors upon request. The ChIP-seq and RNA-seq datasets have been deposited in the Gene Expression Omnibus (GEO) repository with the accession code GSE 95511.

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

## Acknowledgements

This research is supported by the National Research Foundation Singapore and the Singapore Ministry of Education, Singapore Translational Research Award from the Singapore National Medical Research Council (NMRC/STaR/0018/2013 to DGT), National University Hospital System Aspiration Grant (NUHSRO/2014/088/AF-Partner/04 to RT and DGT), and Medical Research Council (MRC, UK) Molecular Haematology Unit grant (MC_UU_12009/6 to TAM, JK, MT and EB). We would like to thank M. Rossner (Ludwig Maximilian University of Munich) for providing *Sharp1* knockout mice, and A. Yokoyama (NCC Tsuruoka Metabolomics Lab) for providing MLL deletion constructs. We thank C. A. Q. Chin (CSI Singapore) for proofreading of the manuscript, H.Q. Hong and G. Chong (CSI Singapore) for making MLL-AF6 construct, R. Thorne (Oxford University) for ChIP-seq data transfer, and members of the Tenen laboratory and CSI Singapore for many helpful discussions. We thank CSI Singapore FACS facility and Duke-NUS Genome Biology Facility for technical support.

## Author contributions

Conceptualization, A.N., A.K., R.T., and D.G.T.; Methodology, A.N., H.S.K., D.B., and R.S. W.; Formal Analysis, A.N., J.L., T.B., and H.Y. Investigation, A.N., H.S.K., J.K., E.B., M.T., Q.Z., and F.L.; Resources, A.K., A.J.D., R.T., and R.D.; Writing—Original Draft, A.N., H.S. K., T.B. and H.Y.; Writing—Review & Editing, R.T. and D.G.T.; Visualization, A.N., J.L., and T.B.; Supervision, H.Y. and D.G.T, Funding Acquisition, T.A.M., R.T., and D.G.T.

## Additional information

**Competing interests:** T.A.M. is one of the founding shareholders of Oxstem Oncology (OSO), a subsidiary company of OxStem Ltd. The remaining authors declare no competing interests.

