## [Peer Review File · Nature Communications]

Reviewers' comments:

Reviewer #1 (Remarks to the Author):

In the manuscript by Numata et al., the authors report SHARP1 as a specific target and essential molecule for MLL-AF6 but not other MLL-rearranged leukemia. Importantly, SHARP1 is largely dispensable for normal hematopoiesis, but its suppression inhibits leukemic cell growth and extends disease latency induced by MLL-AF6. Mechanistically, expression of SHARP1 is regulated by MLL-AF6 and DOT1L. However, immunoprecipitation data also suggests protein-protein interaction and thus potential regulation between SHARP1 and MLL-AF6. While the concept of SHARP1 as a specific target for MLL-AF6 is not novel and has already been reported by Coenen et al., *Leukemia*, 2014, the different conclusion and new mechanistic insights from the current work can be potentially significant and help to advance the field. Nevertheless there are some areas needing further investigation and clarity.

Given the opposite conclusions being made between the current and previous study by Coenen et al., it is essential to have a good account for this key difference: whether SHARP1 dependence is cell line instead of MLL fusion specific (e.g., SHI cells in Coenen's study vs ML2 cells in the current study) or is due to technical reasons. This is important because if SHARP1 dependence is cell line specific, this will have significant implication on the interpretation of the role and therapeutic value of targeting SHARP1 in MLL-AF6 leukemia, the key finding of the current work. Studying and comparing the SHARP1 dependent and independent MLL-AF6 cells can also give mechanistic insights into the role of SHARP1 in MLL-AF6 and the potential resistant mechanisms. Therefore it is critical to include SHI1 cells in the Figures 2-3 to help the authors to make a proper conclusion about the role of SHARP1 in MLL-AF6 leukemia.

In term of the mechanism, more robust biochemical data are needed to support the conclusion that Sharp1 is part of an MLL-AF6 protein complex. The CO-IP with Sharp1 should be performed with other MLL fusions, and also endogenous proteins in appropriate leukemia cell lines. Additionally, the interaction domain on AF6 should be mapped to determine if this also falls on the minimal transformation domain of the fusion, which will further support the role of SHARP1 in MLL-AF6 leukemia. This is another major finding that distinguishes the current study from the previous one, and provides mechanistic insights on how SHARP1 works in MLL-AF6 leukemia.

Reviewer #2 (Remarks to the Author):

In their manuscript "The basic helix-loop-helix transcription factor SHARP1 is an oncogenic driver in MLL-AF6 Acute Myelogenous Leukemia", Tenen and colleagues elucidate mechanisms of MLL-AF6 (MA6) induced transformation that distinguish it from other MLL fusion proteins. Beginning with a combination of CHIP-seq and RNA-expression evaluations of MA6+ cells, the authors show that SHARP1 is one of several unique mediators of MA6's leukemogenic activity and that induction of SHARP1 and a small number of other genes distinguishes MA6 AML from other MLL-fp+ AMLs. The uniqueness of SHARP1 to MA6+ AML is addressed by showing that high levels of SHARP1 mRNA expression data is unique to MA6+ AMLs in several human AML cohorts. The authors convincingly show that SHARP1 contributes to MA6 driven leukemogenesis by performing KD experiments in MA6+ cell lines as well as in an retroviral MA6+ mouse model of AML. Functional studies in vivo demonstrate that MA6-driven AML is significantly attenuated in the absence of SHARP1 due to SHARP1 regulation of LSCs. Intriguingly, SHARP1 binds to many of the same promoters of MA6, highlighting a novel mechanism by which MA6 targets help to reinforce expression of the MA6-driven program.

These studies represent a comprehensive and well-performed set of experiments that convincingly establish that MA6 induces AML through unique mechanisms compared to other MLL fusion proteins. In addition, the manuscript is clearly written and very easy to follow. Major strengths of the paper include: 1) Novel insights regarding the unique mechanisms underlying MA6 induced leukemogenesis, 2) Evaluation of the SHARP1 KO in vivo phenotype, both in normal hematopoiesis as well as in two mouse models of AML; 3) Comprehensive molecular characterization of MA6 and SHARP1 targets to discover their ability to coordinately regulate a unique MA6 AML gene expression program. Weaknesses include: 1) Suboptimal evaluation of the Sharp1^{-/-} normal hematopoietic phenotype; 2) Unconvincing demonstration that SHARP1 induction by MA6 can explain differences in the clinical behavior of MA6 leukemias vs. other types of MLL-fp AMLs. Despite these weaknesses, enthusiasm for this manuscript is very high, and these issues can be addressed relatively easily.

1) The authors show that MA6 induces the expression of unique genes but also share the ability to induce many genes with other MLL fusions, such as the posterior Hox genes. Given that one of the major points of the paper is that MA6 is different from other MLL-fp AML's, the authors should explicitly show the level of induction of the genes that are shared among specific MLL rearranged groups instead of grouping them all together as shown in Suppl. Fig 1.

2) The authors evaluate the SHARP1 KO normal hematopoietic phenotype and claim that there are no differences between WT and KO mice at steady-state. This interpretation is tenuous given the very small number of mice evaluated (n=3/group), as there appears to be trends towards significance between MPP and CLP. As whole BM transplants are likely to mask these effects in lineage potential, the authors should evaluate more mice and consider performing clonal in vitro assays of lymphoid potential to characterize these differences (in vivo experiments would not be necessary).

3) Authors show MLLN CHIP-seq data in the SHARP1 locus in MA6+ AML cells in order to make the argument that SHARP1 is uniquely regulated by MA6, but they do not show a similar experiment in which MLLN-CHIP was performed in a non-MA6+ MLL-fp+ AML cell. This would be required to establish that differences in SHARP1 mRNA expression are not mediated at the post-transcriptional level.

4) The authors argue that induction of SHARP1 by MA6 and its ability to regulate MA6 target genes can explain why MA6+ patients do worse than other MLL-FP patients. While they have convincingly shown that MA6 likely induces AML through unique mechanisms, it is not clear this is the reason for this group's poorer clinical behavior. In fact, even the retroviral overexpression models show that MA9+ AMLs induce death faster. The authors are urged to adjust their discussion to reflect that this specific issue has not been resolved (although this can now be proposed as a possible mechanism for the differences in clinical behavior).

5) The authors attempt to show that the role of SHARP1 is unique to MA6 AMLs by comparing survival between SHARP1 WT and KO mice using the MA6 and MA9 models. While the fact that they have performed this experiment is impressive and very much appreciated, the MA9 mouse model is not a very good one for parsing out subtle differences that might exist between WT and KO mice. In my view, a more helpful experiment would be to perform SHARP1 shRNA experiments in non-MA6 MLL-fp cell lines to demonstrate that this does not result in a change in apoptosis/growth as already shown in MA6+ cells.

MINOR

1) The authors describe SHARP1 mRNA expression in two cohorts of AML (Figure 1D). A few comments: 1) these figures should be increased in size because they are very difficult to evaluate; 2) The authors should show/describe how much SHARP1 expression is increased in the MA6 cases in these two cohorts; 3) The authors should show a direct comparison of SHARP1 expression in these cohorts vs other MLLr cases as shown in Fig 1C. The authors should also address whether their SHARP1 target genes are also upregulated in these AML patient samples to tie their findings back to human AML.

2) Authors show that SHARP1 KD in MA6+ human AML cell lines induces apoptosis (Fig 3). Is this

associated with differentiation or is this a pure survival phenotype?

3) "rarely detected" (Line 121) is not appropriate to describe levels of expression. Please change to "decreased"

4) Line 182. Authors state that reduced WBC counts and increased RBCs reflect "delayed leukemic transformation". Formally, the authors cannot claim this since transformation refers to the time to appearance of the first LSC. Should be changed to "decreased disease aggressiveness" or something similar.

5) Authors show cytology of cells in the liver and spleen in Fig 4d. There appears to be evidence of significant differentiation in the images shown, raising the possibility this is more like an MPN. Full differential with blast counts or some other method of enumerating bona fide blasts should be shown.

6) Related to this issue, what cells are being transplanted in primary and secondary transplants following retroviral MA6 transduction? If equal numbers of LSCs or blasts were not transplanted, this should be mentioned to provide a more meaningful understanding of the differences in survival between the WT and KO transduced cells.

7) The authors mention the cytoplasmic function of AF6 in their introduction but never return to this issue experimentally or in the discussion. Given this, perhaps this is not an important point to make in the introduction?

Reply to the Referees

We thank the referees for the efforts to evaluate and improve the manuscript. The referees' comments and respective additions/corrections have significantly strengthened the study. We made substantial changes as requested. For the ease of the reviewers, we first outline all of our changes, followed by a detailed point-by-point response. With this extensive revision, we believe that the manuscript has been substantially strengthened.

Outline of changes:

1. We have re-evaluated the expression level of the common MLL-FP target genes (*HOXA9*, *HOXA10*, and *MEIS1*) among specific MLLr-AML subtypes (for Reviewer #2)
2. We have re-evaluated the SHARP1 expression level in the two cohorts and made a comparison (for Reviewer #2).
3. We have extended shRNA-mediated SHARP1 knockdown studies to another MLL-AF6 AML cell line, SHI-1, to resolve any possible differences between the previous study and the ours (for Reviewer #1).
 - a) quantitative PCR for SHARP1
 - b) Genome-wide MLL-N ChIP-seq analysis (new ChIP-seq tracks are in the process of deposition in GSE 95511)
 - c) SHARP1 knockdown studies (qPCR, WB, apoptosis assay, cell count, CFC assay)
4. We have extended the analysis of MLL-N and H3K79me2 ChIP-seq to non-MLL-AF6 MLLr-AML cell lines, MV4-11 (MLL-AF4) and THP-1 (MLL-AF9) (for Reviewer #2).
5. We have assessed the differentiation status of MLL-AF6 human AML cell lines upon SHARP1 knockdown (for Reviewer #2).
6. We have extended the shRNA-mediated SHARP1 knockdown studies to non-MLL-AF6 cell lines, MOLM-14 (MLL-AF9) and MV4-11 (MLL-AF4) (apoptosis assay, cell count, CFC assay) (for Reviewer #2)
7. We have assessed the differentiation status of mouse MLL-AF6 leukemic BM cells (for Reviewer #2).
8. We have re-evaluated the hematopoietic phenotype of SHARP1 knockout mice (for Reviewer #2).
9. We have expanded on the study of interaction between MLL-fusion proteins and SHARP1 protein (for Reviewer #1)
 - a) Co-immunoprecipitation data on SHARP1 interaction with MLL-AF9
 - b) MLL domain mapping for identification of the region for SHARP1 interaction
 - c) Attempts to detect endogenous interaction between MLL-AF6 and SHARP1 protein
10. We have evaluated the expression level of SHARP1 target genes in AML patients (for Reviewer #2)

Reviewer #1

In the manuscript by Numata et al., the authors report SHARP1 as a specific target and essential molecule for MLL-AF6 but not other MLL-rearranged leukemia. Importantly, SHARP1 is largely dispensable for normal hematopoiesis, but its suppression inhibits leukemic cell growth and extends disease latency induced by MLL-AF6. Mechanistically, expression of SHARP1 is regulated by MLL-AF6 and DOT1L. However, immunoprecipitation data also suggests protein-protein interaction and thus potential regulation between SHARP1 and MLL-AF6. While the concept of SHARP1 as a specific target for MLL-AF6 is not novel and has already been reported by Coenen et al., Leukemia, 2014, the different conclusion and new mechanistic insights from the current work can be potentially significant and help to advance the field. Nevertheless there are some areas needing further investigation and clarity.

Comment 1. Given the opposite conclusions being made between the current and previous study by Coenen et al., it is essential to have a good account for this key difference: whether SHARP1 dependence is cell line instead of MLL fusion specific (e.g., SHI cells in Coenen's study vs ML2 cells in the current study) or is due to technical reasons. This is important because if SHARP1 dependence is cell line specific, this will have significant implication on the interpretation of the role and therapeutic value of targeting SHARP1 in MLL-AF6 leukemia, the key finding of the current work. Studying and comparing the SHARP1 dependent and independent MLL-AF6 cells can also give mechanistic insights into the role of SHARP1 in MLL-AF6 and the potential resistant mechanisms. Therefore it is critical to include SHI1 cells in the Figures 2-3 to help the authors to make a proper conclusion about the role of SHARP1 in MLL-AF6 leukemia.

Response:

SHI-1 cells express high SHARP1 (Figure 2A of the revised manuscript), and MLL^N binds to the *SHARP1* gene locus as well as posterior *HoxA* gene cluster in ChIP-seq analysis of SHI-1 cells (Figure 2C), suggesting SHARP1 is transcribed directly by MLL-AF6. We proceeded with knockdown experiments using two independent SHARP1 shRNAs, which induced substantial reduction of SHARP1 in RNA and protein (Figures 3A and S3A). Consistent with the findings in the other two MLL-AF6 cell lines, ML-2 and CTS, SHI-1 cells demonstrated robust apoptosis, reduced cell growth in culture, and reduced activity in CFC assays. Since we obtained consistent results from three out of three MLL-AF6 AML cell lines, we conclude that SHARP1 dependence is not cell line-specific. We have now moved the CTS data from a supplemental figure to the main figures, and show the new data together in Figures 2, 3, and S3A as the reviewer suggested. We added new descriptions for SHI-1 in the part of description of the use cell lines and moved it to “Results” section under “MLL-AF6 Directly Upregulates SHARP1 by DOT1L” sub-section. We discuss differences between Coenen’s study and ours in our revised “Discussion” as follows:

Figure legend for Figures 2A and 2C

(A) SHARP1 mRNA expression in human AML cell lines assessed by real time PCR. The cell lines analyzed are: ML-2 (MLL-AF6), CTS (MLL-AF6), SHI-1 (MLL-AF6), MOLM-14 (MLL-AF9), MV4-11 (MLL-AF4) and Kasumi-1 (AML1-ETO).

(C) ChIP-seq profiles of SHI-1 cells using MLL^N antibody and input at the loci of *HOXA* gene cluster (left) and *SHARP1* (right).

Line 124

“In human AML cell lines, consistent with our findings in the gene expression profiles from the multiple AML cohorts, SHARP1 mRNA was expressed highly in ML-2, CTS and SHI-1 cells, all of which harbor t(6;11)(q27;q23), whereas it was undetectable in MOLM-14, MV4-11, and Kasumi-1, which harbor t(9;11)(p22;q23), t(4;11)(q21;q23), and t(8;21)(q22;q22), respectively (Figure 2A).”

Line 142

“To confirm these findings in another MLL-AF6 AML cell line, we performed an independent ChIP-seq analysis of SHI-1 cells which expresses both MLL and MLL-AF6, demonstrating that MLL^N binds to *SHARP1* gene loci as well as posterior *HOXA* genes locus (Figure 2C).

Line 401

“However, a recent study by Coenen et al. demonstrated that shRNA-mediated SHARP1 knockdown did not have any effect in SHI-1 cells⁵³, in contrast to our findings. This discrepancy might be explained by the difference in knockdown efficiencies with the use of different shRNAs against SHARP1. In fact, one of the shRNAs was common between their study and ours, and has led to reduced growth and increased apoptosis in SHI-1 cells, even though the differences were only significant in our study. Based on our findings in the three MLL-AF6 and two other MLLr-AML cell lines, as well as genetic deletion in murine AML models, we concluded that SHARP1 plays an oncogenic role in MLL-AF6 AML cells.”

Comment 2. In term of the mechanism, more robust biochemical data are needed to support the conclusion that Sharp1 is part of an MLL-AF6 protein complex. The CO-IP with Sharp1 should be performed with other MLL fusions, and also endogenous proteins in appropriate leukemia cell lines.

Additionally, the interaction domain on AF6 should be mapped to determine if this also falls on the minimal transformation domain of the fusion, which will further support the role of SHARP1 in MLL-AF6 leukemia. This is another major finding that distinguishes the current study from the previous one, and provides mechanistic insights on how SHARP1 works in MLL-AF6 leukemia.

Response:

We attempted endogenous co-immunoprecipitation of SHARP1 and MLL-AF6 using nuclear extracts from ML-2 and CTS cells (see data for reviewers only #1). Despite numerous attempts, using a variety of protocols, we could not detect robust, consistent interactions between the two endogenous proteins convincing enough for

publication. We believe that this may be largely because SHARP1 has a molecular mass similar to that of heavy chain, and heavy chain bands may have obscured the SHARP1 bands. As an alternative approach, we performed proximity ligation assay (PLA) and detected interaction between SHARP1 and MEN1 (#2), one of the key co-factors within the MLL-FP complex. Our PLA results support the hypothesis that SHARP1 interacts with either MLL-AF6 and/or the MLL-AF6 protein complex to regulate MLL-AF6 target genes. We have included these data for the reviewers as following:

Data for reviewers only #1. Co-immunoprecipitation studies of SHARP1 and MLL-AF6.

Left: Nuclear extracts from SHI-1 cells were subjected to immunoprecipitation (IP) with anti-MLL^N antibody. Proteins present in IP (lane 2 and 3) or nuclear extract (lane 1, input) were separated by SDS-PAGE and immunoblotted with antibodies specific for MLL^N (top panel), MEN1 (middle panel) and SHARP1 (bottom panel). We were unable to clearly detect MLL-AF6 as SHI-1 cells have low expression level (lane 1). Detection of interaction between MLL-AF6 and MEN1, a key protein in MLL-AF6 protein complex, compared to the control IP with IgG antibody, confirms MLL-AF6 IP (lane 2 and 3).

Right: Nuclear extracts from ML-2 cells were subjected to IP with anti-AF6 and anti-SHARP1 antibodies. Proteins present in IP (lane 2, 3 and 4) or nuclear extract (lane 1) were separated by SDS-PAGE and immunoblotted with antibodies specific for AF6 (top panel) and SHARP1 (bottom panel). IP with anti-SHARP1 antibody suggests interaction of SHARP1 and MLL-AF6, compared to the control IP with IgG antibody (lane 2 and 4). IP with anti-AF6 antibody demonstrates IP of MLL-AF6 (lane 3). Multiple ladder-like bands in MLL-AF6 might correspond to protein degradation (indicated by asterisks). SHARP1 signal, which was obscured by heavy chain band, cannot be clearly detected in IP in both SHI-1 and ML-2 cells. Molecular sizes of protein markers are shown on the left.

Data for reviewers only #2. Proximity ligation assay (PLA) shows interaction of SHARP1 and MEN1 in ML-2 cells. Nuclei were stained with DAPI (in blue). PLA detection of SHARP1 and MEN1 interaction are observed as distinct fluorescent foci (in red). Overlaid image indicates that the interactions are localized in the nuclei.

As suggested, we also performed co-IP with SHARP1 and MLL-AF9 constructs overexpressed in 293T cells and observed a robust interaction. To determine whether SHARP1 interacts with the MLL portion that is shared in both MLL fusion proteins (MLL-FPs), we carried out domain mapping to identify the specific regions within MLL responsible for SHARP1 interaction. Indeed, we demonstrated that SHARP1 interacts within MLL 541 – 1251aa, which has been reported to have a role in recruiting epigenetic modifiers. Although this suggested that SHARP1 interacts with MLL-FPs in general, we concluded that this interaction is observed only in MLL-AF6 AML cells due to SHARP1 specific expression in MLL-AF6. We have included these data in Figures 7F, 7G and S7A, and based on these new findings we modified the subtitle of the paragraph and added the new schematic depicting our findings in Figure 8 so that the readers could understand our proposed model clearly.

Figure legend for Figures 7F and 7G

(F) Schematic showing a series of MLL deletion mutants and the MLL interaction regions with SHARP1. Interaction is indicated by + sign and loss of interaction by – sign. Shaded boxes indicate AT hook motifs, nuclear translocation sequences (NTS1 and NTS2), subnuclear localization domains (SNL1 and SNL2) and methyltransferase domain (MT).

(G) Domain mapping analysis of MLL required for interaction with SHARP1. 293T cells are transfected with plasmids encoding FLAG-tagged MLL truncation mutants and HA-tagged SHARP1. Whole cell lysates were prepared from the transfected cells and subjected to immunoprecipitation with anti-FLAG antibody. Proteins present in immunoprecipitates (IP, lane 1–7) or whole cell lysates of transfected cells (input, lane 8–14) were separated by SDS-PAGE and immunoblotted with antibodies specific for FLAG-tagged MLL truncation mutants and HA-tagged SHARP1. Interaction of SHARP1 and MLL truncation mutants was detected in lane 2, 3, 5, 6 and 7, indicating that amino acids 541–1251 of MLL is required for interaction with SHARP1.

Figure legend for Figure S7A

(A) Co-immunoprecipitation studies of SHARP1 and MLL-AF9 with an anti-HA antibody in 293T cells transfected with plasmids encoding MLL-AF9 and/or HA-tagged SHARP1. Proteins present in immunoprecipitates (IP, lane 1-4) or whole cell lysates of transfected cells (input, lane 5-8) were separated by SDS-PAGE and immunoblotted with antibodies specific for MLL^N and SHARP1. Interaction of SHARP1 and MLL-AF9 was detected (lane 4) and not observed in negative control lanes with either empty vector, SHARP1 or MLL-AF9 only (lane 1-3).

Figure legend for Figure 8

Figure 8. Oncogenic Role for SHARP1 in MLL-AF6 AML

Unlike other MLL fusion proteins, MLL-AF6 protein binds to and activates *SHARP1* gene in a DOT1L-dependent manner. Upregulated SHARP1 binds to E-box motifs in active chromatin, and also interacts with MLL-AF6 to regulate a subset of genes critical for leukemogenicity. This unique transcriptional machinery contributes to the maintenance of MLL-AF6 leukemic stem cells.

Line 320

Original subtitle: “SHARP1 Regulates MLL-AF6 Target Genes”

Revised subtitle: “SHARP1 Cooperates with MLL-AF6 to Regulate Target Genes”

Line 335

“To determine whether SHARP1 exists in a complex with MLL-AF6, we carried co-immunoprecipitation (co-IP) experiments in nuclear extracts from MLL-AF6 cell lines ML-2 and SHI-1. AF6 or MLL^N co-IPs failed to detect SHARP1, which may have been obscured by heavy chain bands (data not shown). To circumvent this issue, we performed co-IP experiments in 293T cells that were transfected with MLL-AF6 and HA-tagged SHARP1, and demonstrated a robust interaction between the two proteins (Figure 7E). Intriguingly, we also observed interaction between SHARP1 and MLL-AF9 (Figure S7A), indicating that SHARP1 interacts with the MLL portion that is present in both MLL fusions. Indeed, using a series of MLL deletion mutants¹⁵, we identified a region, amino acids 541-1251 (541-1251aa) of MLL, which was responsible for interaction with SHARP1. We did not observe an interaction with MLL (1-540aa), while the interactions with other MLL mutants were comparable to that of MLL (1-1251aa) (Figures 7F and 7G), indicating that SHARP1 interaction with MLL-AF6 and MLL-AF9 is dependent on MLL (541-1251aa). This region contains the transcriptional repression domain (RD1 and RD2) (1101-1400aa), including a methyltransferase domain that shares homology to methyl DNA-binding proteins^{41,42} and is known for the recruitment of repressor complexes containing HDAC1⁴². Given these findings, it is conceivable that the interaction with SHARP1 could alter the constituents of the MLL-AF6 complex and influence the regulation of target genes. Although SHARP1 interacts with common portion of MLL-FP, its specific expression in MLL-AF6 only might provide a unique mechanism in regulation of the MLL-AF6 target genes.”

New references

41. Cross, S.H., Meehan, R.R., Nan, X. & Bird, A. A component of the transcriptional repressor MeCP1 shares a motif with DNA methyltransferase and HRX proteins. *Nat Genet* **16**, 256-9 (1997).
42. Xia, Z.B., Anderson, M., Diaz, M.O. & Zeleznik-Le, N.J. MLL repression domain interacts with histone deacetylases, the polycomb group proteins HPC2 and BMI-1, and the corepressor C-terminal-binding protein. *Proc Natl Acad Sci U S A* **100**, 8342-7 (2003).

Line 362; addition of“(Figure 8)”

We demonstrated that SHARP1 plays an oncogenic role to maintain the clonogenic ability and leukemia-initiating potential, regulating the expression of genes crucial for leukemia cell survival including target genes of MLL-AF6 (Figure 8).

Reviewer #2

In their manuscript The basic helix-loop-helix transcription factor SHARP1 is an oncogenic driver in MLL-AF6 Acute Myelogenous Leukemia , Tenen and colleagues elucidate mechanisms of MLL-AF6 (MA6) induced transformation that distinguish it from other MLL fusion proteins. Beginning with a combination of CHIP-seq and RNA-expression evaluations of MA6+ cells, the authors show that SHARP1 is one of several unique mediators of MA6's leukemogenic activity and that induction of SHARP1 and a small number of other genes distinguishes MA6 AML from other MLL-fp_+ AMLs. The uniqueness of SHARP1 to MA6+ AML is addressed by showing that high levels of SHARP1 mRNA expression data is unique to MA6+ AMLs in several human AML cohorts. The authors convincingly show that SHARP1 contributes to MA6 driven leukemogenesis by performing KD experiences in MA6+ cell lines as well as in an retroviral MA6+ mouse model of AML. Functional studies in vivo demonstrate that MA6-driven AML is significantly attenuated in the absence of SHARP1 due to SHARP1 regulation of LSCs. Intriguingly, SHARP1 binds to many of the same promoters of MA6, highlighting a novel mechanism by which MA6 targets help to reinforce expression of the MA6-driven program.

These studies represent a comprehensive and well-performed set of experiments that convincingly establish that MA6 induces AML through unique mechanisms compared to other MLL fusion proteins. In addition, the manuscript is clearly written and very easy to follow. Major strengths of the paper include: 1) Novel insights regarding the unique mechanisms underlying MA6 induced leukemogenesis, 2) Evaluation of the SHARP1 KO in vivo phenotype, both in normal hematopoiesis as well as in two mouse models of AML; 3) Comprehensive molecular characterization of MA6 and SHARP1 targets to discover their ability to coordinately regulate a unique MA6 AML gene expression program. Weaknesses include: 1) Suboptimal evaluation of the Sharp1-/- normal hematopoietic phenotype; 2) Unconvincing demonstration that SHARP1 induction by MA6 can explain differences in the clinical behavior of MA6 leukemias vs. other types of MLL-fp AMLs. Despite these weaknesses, enthusiasm for this manuscript is very high, and these issues can be addressed relatively easily.

Comment 1. The authors show that MA6 induces the expression of unique genes but also share the ability to induce many genes with other MLL fusions, such as the posterior Hox genes. Given that one of the major points of the paper is that MA6 is different from other MLL-fp AML's, the authors should explicitly show the level of induction of the genes that are shared among specific MLL rearranged groups instead of grouping them all together as shown in Suppl. Fig 1.

Response:

We found that expression levels of canonical target genes of MLL-FPs (*HOXA9*, *HOXA10*, and *MEIS1*) in MLL-AF6 are not different from those in the specific subtypes of MLLr-AML (MLL-AF9, MLL-AF10, MLL-ENL, MLL-ELL, MLL-SEPTIN6, MLL-AF4, ALL-AF1q) except for two comparisons, *HOXA9* between MLL-AF6 vs MLL-SEPTIN6, and *HOXA10* between MLL-AF6 vs MLL-AF9. We modified the descriptions for common target genes expressions in Results" section under "SHARP1 is Overexpressed in MLL-AF6 AML" sub-section.

Among these genes, we identified nine MLL-AF6 targets (*SHARPI*, *P2RY1*, *TRPS1*, *SSPN*, *FAM169A*, *MMRN1*, *SKIDA1*, *HOXA7*, and *SLC35D1*) (Figures 1B, 1C, S1A and Table 1), whereas there was no difference in the expression level of canonical targets of MLL-FPs (*HOXA9*, *HOXA10*, and *MEIS1*) between MLL-AF6 and the specific subtypes generally (Figure S1B).

Comment 2. The authors evaluate the SHARPI KO normal hematopoietic phenotype and claim that there are no differences between WT and KO mice at steady-state. This interpretation is tenuous given the very small number of mice evaluated (n=3/group), as there appears to be trends towards significance between MPP and CLP. As whole BM transplants are likely to mask these effects in lineage potential, the authors should evaluate more mice and consider performing clonal in vitro assays of lymphoid potential to characterize these differences (in vivo experiments would not be necessary).

Response:

We have evaluated more mice (n = 5 per group) and confirmed that there was no significant difference in a cell number of MPP between *SHARPI* KO and WT mice. For analysis of CLP, we performed more accurate and stringent gating based on past literature (Kondo *et al.*, Cell 1997, 91: 661, Karsunky *et al.*, Blood 2008, 111: 5562), and we concluded that there was no significant difference in a cell number of CLP between the two groups. Given these convincing *in vivo* results, we did not perform *in vitro* progenitor assay. We have revised Figures 5C and S5B and the relevant description in the “Results” section under “Sharp1 is Dispensable for Steady State Hematopoiesis” sub-section accordingly.

Reference

Kondo *et al.*, Identification of clonogenic common lymphoid progenitors in mouse bone marrow. Cell, **91**: 661-72 (1997)

Karsunky *et al.*, Flk2+ common lymphoid progenitors possess equivalent differentiation potential for the B and T lineages. Blood, **111**: 5562-70 (2008)

Line 249

We did not find any differences in the number of hematopoietic stem cell (HSC; CD150⁺CD48⁻LSK) and progenitor populations (MPP = multipotent progenitor CD150⁻CD48⁺LSK; CMP = common myeloid progenitor Lin⁻c-kit⁺Sca1⁻CD34⁺CD16/32⁻; GMP = Lin⁻c-kit⁺Sca1⁻CD34⁺CD16/32⁺; MEP = myeloid erythroid progenitor Lin⁻c-kit⁺Sca1⁻CD34⁻CD16/32⁻; CLP = common lymphoid progenitor, Lin⁻IL7R⁺c-kit⁺Sca1⁺Flk2⁺) between *Sharp1*^{+/+} mice and *Sharp1*^{-/-} mice. The frequency of mature granulocytes (Gr1⁺CD11b⁺) and B cells (B220⁺) also did not demonstrate any difference (Figure 5C and S5B).

Comment 3. Authors show MLLN CHIP-seq data in the SHARPI locus in MA6+ AML cells in order to make the argument that SHARPI is uniquely regulated by MA6, but they do not show a similar experiment in which MLLN-CHIP was performed in a non-MA6+ MLL-fp+ AML cell. This would be required to establish that differences in SHARPI mRNA expression are not mediated at the post-transcriptional level

Response:

We have performed MLL^N and H3K79me2 ChIP-seq analysis in THP-1 (MLL-AF9) and MV4-11 (MLL-AF4) cells. We demonstrated that the *SHARP1* locus is neither bound by MLL-AF9 or MLL-AF4, nor enriched with H3K79me2. Given these findings, we concluded that SHARP1 expression is not suppressed at the post-transcriptional level in the other subtypes of MLLr-AML. We have included these data in Supplemental Figure 2B. The following sentences were added in the “Results” section under “MLL-AF6 Directly Upregulates SHARP1 by DOT1L” sub-section and the redundant descriptions in the “Results” and Table S3 were removed.

Figure legend for Supplemental Figure S2B

(B) MLL^N ChIP-seq profiles of THP-1 (MLL-AF9) (top panel) and MV4-11 cells (MLL-AF4) (bottom panel) at the loci of the *HOXA* gene cluster and *SHARP1* gene.

Line 145

“To ascertain the unique MLL-AF6 binding, we analyzed MLL^N and H3K79me2 ChIP-seq data of THP-1 (MLL-AF9) and MV4-11 (MLL-AF4) cells and found that neither MLL^N binding nor H3K79me2 enrichment was observed at *SHARP1* loci (Figure S2B). Collectively, our results indicate that *SHARP1* is a unique transcriptional target of MLL-AF6 and its expression is not suppressed at the post-transcriptional level in the other subtypes of MLLr-AML.”

Removed sentences (originally in page 4)

We further analyzed their expression in non-MLLr-AMLs and found that five genes (*SHARP1*, *P2RY1*, *TRPS1*, *SSPN*, and *FAM169A*) demonstrated specific upregulation in MLL-AF6 (Figure S1A and Table S3).

Importantly, all five genes were neither bound by MLL-AF9, MLL-AF4 nor MLL-ENL based on ChIP-seq analysis from previous studies^{8,20,21}.

Comment 4. The authors argue that induction of SHARP1 by MA6 and its ability to regulate MA6 target genes can explain why MA6+ patients do worse than other MLL-FP patients. While they have convincingly shown that MA6 likely induces AML through unique mechanisms, it is not clear this is the reason for this group's poorer clinical behavior. In fact, even the retroviral overexpression models show that MA9+ AMLs induce death faster. The authors are urged to adjust their discussion to reflect that this specific issue has not been resolved (although this can now be proposed as a possible mechanism for the differences in clinical behavior).

~~~~~

**Response:**

We agree with the reviewer that SHARP1 overexpression, although demonstrated as a unique mechanism for MLL-AF6 leukemogenesis, was insufficient to explain for the poorer clinical behavior. As such, we have adjusted our discussion.

Line 356

Original sentence

“Our study revealed a mechanism to explain the distinct features of MLL-AF6 AML, which have a dismal prognosis compared to other MLL fusion AMLs.”

Revised sentence

“Our study reveals a unique mechanism in the leukemogenicity in MLL-AF6 AML.”

*Comment 5. The authors attempt to show that the role of SHARP1 is unique to MA6 AMLs by comparing survival between SHARP1 WT and KO mice using the MA6 and MA9 models. While the fact that they have performed this experiment is impressive and very much appreciated, the MA9 mouse model is not a very good for parsing out subtle differences that might exist between WT and KO mice. In my view, a more helpful experiment would be to perform SHARP1 shRNA experiments in non-MA6 MLL-fp cell lines to demonstrate that this does not result in a change in apoptosis/growth as already shown in MA6+ cells.*

~~~~~

Response:

We have performed SHARP1 shRNA mediated knockdown experiments in MOLM-14 (MLL-AF9) and MV4-11 (MLL-AF4) AML cell lines. We confirmed that two SHARP1 shRNAs neither induced apoptosis nor suppressed cell growth. We have now included these data in Supplemental Figures 3C, 3D and 3E. Accordingly, the following line has been added in the second paragraph in the “Results” section under “SHARP1 Maintains the Growth and Clonogenic Ability of MLL-AF6 Cells” sub-section:

Figure legend for Figures S3C, S3D and S3E

(C) Representative AnnexinV and PI FACS plot and percentage of AnnexinV⁺ and PI⁻ cell of MOLM-14 cells (top panel) and MV4-11 cells (bottom panel) transduced with the indicated shRNAs.

(D) Cell count of MOLM-14 cells (left panel) and MV4-11 cells (right panel) transduced with the indicated shRNAs in culture. The value is determined as fold increase in cell number relative to the number of cells initially plated.

(E) Colony-forming units (CFU) per 10,000 cells of MOLM-14 cells (left panel) and MV4-11 cells (right panel) transduced with the indicated shRNAs, with the number of colonies observed 7 days after the plating. Error bars represent \pm SEM.

Line 176

“However, transduction of the two SHARP1 shRNA neither induced apoptosis nor attenuated cell growth and colony-forming ability in MOLM-14 (MLL-AF9) and MV4-11 (MLL-AF4) (Figures S3C-E).”

Comment 6. The authors describe SHARP1 mRNA expression in two cohorts of AML (Figure 1D). A few comments: 1) these figures should be increased in size because they are very difficult to evaluate 2) The authors should show/describe how much SHARP1 expression is increased in the MA6 cases in these two cohorts 3) The authors should show a direct comparison of SHARP1 expression in these cohorts vs other MLLr cases as shown in Fig 1C. The authors should also address whether their SHARP1 target genes are also upregulated in these AML patient samples to tie their findings back to human AML.

Response:

1)2)3) We have enlarged the size of Figures 1D and 1E, added the graphs to show the SHARP1 expression levels in MLL-AF6 cases in comparison to other MLLr and non-MLLr, and described the expression values in the figure legends. We also showed SHARP1 expression values in the figure legend for Figure 1C.

Figure legend for 1C, 1D and 1E

(C) Box plot showing SHARP1 log₂ expression level in AML patients and normal bone marrow (NBM) CD34+ cells. SHARP1 log₂ expression level: MLL-AF6: 7.504±0.788 (n=14), Other MLL: 2.854±0.065 (n=42), Non-MLL: 3.623±0.064 (n=276), NBM CD34: 2.856±0.036 (n=12).

(D and E) Left panel: Unsupervised hierarchical gene-expression clustering from 2 distinct cohorts (D; n=285 and E; n=268) of adult AML patients. The bars indicate *SHARP1* mRNA expression. All of the five high *SHARP1* cases have the MLL-AF6 fusion gene. Right panel: Box plot showing SHARP1 log₂ expression in AML patients. SHARP1 log₂ expression level: (D) MLL-AF6: 10.45±0.096 (n=3), Other MLL: 4.383±0.082 (n=15), Non-MLL: 4.566±0.045 (n=258), (E) MLL-AF6: 11.18±0.828 (n=2), Other MLL: 5.916±0.257 (n=16), Non-MLL: 5.658±0.033 (n=241). Error bars represent ± SEM. ***p < 0.001

3) To address the reviewer's question regarding common genes between SHARP1 target genes and upregulated genes in MLL-AF6+ AML patient samples, we compared the 319 SHARP1 target genes that we identified using an integrative analysis of RNA-seq and ChIP-seq in ML-2 cells, and the 581 genes expressed higher in MLL-AF6 AML cases than other MLLr-AML, and identified 17 genes that overlap. In the manuscript and figures, we demonstrated several SHARP1 targets and SHARP1/MLL-AF6 co-targets which are relevant to leukemogenicity and cancer, however, we did not identify them in the 17 overlapping genes. We described the lack of overlap and put forward a plausible explanation for this. The following paragraph was added to the "Discussion" section and the new figure of ChIP-seq analysis was added as a Supplemental Figure 7B, which indicated that SHARP1 binding sequences in the gene promoter could be bound by other transcription factors.

Line 427

"We identified a subset of (a) SHARP1 targets and (b) co-targets of MLL-AF6 and SHARP-1 that are critical for leukemogenicity using an integrative analysis of RNA-seq and ChIP-seq datasets in ML-2 cells. However, these genes are not overexpressed in MLL-AF6 AML patients compared to the other subtypes of MLLr-AML (data not shown). SHARP1 ChIP-seq analysis highlighted that various motifs of potential co-factors are enriched in the promoter of SHARP1 targets (Figure S7B), suggesting a more complex mechanism of regulation involving other transcription factors. It is plausible that those genes are activated by other transcription factors in different subtypes of AML that generally do not express SHARP1. It will be of future interest to investigate how overexpressed SHARP1 influences the recruitment of transcriptional regulatory factors to chromatin, providing a unique mechanism for gene regulation in ML-AF6 AML."

Figure legend for Figure S7B

(B) Top 3 enriched motifs within SHARP1 ChIP-seq peaks on gene promoters revealed by peak-motifs module from the RSAT suite, using oligomer length ranging from 6 to 8 nucleotides and the “merge lengths for assembly” option.

Comment 7. Authors show that SHARP1 KD in MA6+ human AML cell lines induces apoptosis (Fig 3). Is this associated with differentiation or is this a pure survival phenotype?

Response:

We have assessed the differentiation status of ML-2 and CTS cells upon SHARP1 KD by FACS and Giemsa staining cytology. We did not find significant characteristics of granulocytic or monocyte differentiation in both ML-2 and CTS cell lines, which indicated that SHARP1 KD induced pure apoptosis. The new figures were added in Supplemental Figure 3 and the following sentence was added to the “Results” section under "SHARP1 Maintains the Growth and Clonogenic Ability of MLL-AF6 AML Cells" subsection.

Figure legend for Figure S3B

(B) Expression levels of mature granulocytic and monocytic markers, CD11b, CD14 and CD15, measured by FACS in ML-2 and CTS cells transduced with the indicated shRNAs (top panel). Giemsa staining of ML-2 and CTS cells transduced with the indicated shRNAs (bottom panel).

Line 172

“Consistently, downregulation of SHARP1 increased apoptotic cells (AnnexinV⁺ DAPI⁻ or PI⁻) (Figure 3C), while granulocytic and monocytic differentiation was not observed assessed by both flow cytometry and morphological analysis (Figure S3B). We also observed attenuated cell growth (Figure 3D) and colony formation (Figure 3E).”

Comment 8. rarely detected (Line 121) is not appropriate to describe levels of expression. Please change to decreased.

Response:

We agree with the reviewer on this point. We have changed the referred sentence to the following:

Line 110

“Importantly, SHARP1 was decreased in most cases of other subtypes of AML as well as normal bone marrow (BM) CD34-positive cells (Figure 1C).

Comment 9. Line 182. Authors state that reduced WBC counts and increased RBCs reflect delayed leukemic transformation. Formally, the authors cannot claim this since transformation refers to the time to appearance of the first LSC. Should be changed to decreased disease aggressiveness or something similar.

Response: We agree with the reviewer on this point. We have changed the sentence (Line 182) to the following and we also changed the subtitle accordingly.

Line 181

Original subtitle “*Deletion of Sharp1 Delays Leukemic Transformation of MLL-AF6 AML*”

Revised subtitle “*Deletion of Sharp1 Attenuates MLL-AF6 AML Progression*”

Line 191

“Recipients of MA6/WT demonstrated higher white blood cell (WBC) counts (median 26.5 vs 7.18 x 10³/μL, $p = 0.0004$) and lower red blood cell (RBC) counts (median 6.51 vs 8.74 x 10⁶/μL, $p = 0.0097$), as compared to MA6/S1KO, demonstrating that deletion of SHARP1 decreased disease aggressiveness.”

Comment 10. Authors show cytology of cells in the liver and spleen in Fig 4d. There appears to be evidence of significant differentiation in the images shown, raising the possibility this is more like an MPN. Full differential with blast counts or some other method of enumerating bona fide blasts should be shown.

Response:

We apologize for the poor annotation of Figure 4D. The cytology panels were generated from bone marrow cells. To clarify this better, labels are added next to the pictures. A full differential blast count has been performed to confirm that there was no significant difference in differentiation status. Although mature myeloid cells were shown in the photos, myeloblasts accounted for approximately 65% of all nucleated cells in the bone marrow from the both groups, meeting the criteria of AML. The following sentences were added in the Results section under “*Deletion of Sharp1 Attenuates MLL-AF6 AML Progression*” subsection:

Figure legend for Figure 4D and 4E

(D) Pictures of liver and spleen and Wright Giemsa staining of BM cells from moribund leukemic mice.

(E) Left panel: Differential count of Wright Giemsa-stained BM cells from 3 independent moribund leukemic mice. Mybl denotes myeloblasts, myelo denotes promyelocytes, myelocytes and metamyelocytes, band (seg) denotes band and segmented neutrophils, and others denote lymphocytes and macrophages. Error bars represent \pm SEM.

Line 195

“Moribund recipients from both groups displayed enlargement of liver and spleen (Figure 4D). The majority of the bone marrow cells were immature Gr1 and CD11b positive myeloblasts (Figure 4D, 4E and S4A) and did not present any difference in the differentiation status between the two groups (Figure 4E).

Comment 11. Related to this issue, what cells are being transplanted in primary and secondary transplants following retroviral MA6 transduction? If equal numbers of LSCs or blasts were not transplanted, this should be mentioned to provide a more meaningful understanding of the differences in survival between the WT and KO transduced cells.

Response:

In primary transplants, equal numbers of transduced LSK cells were transplanted, whereas in secondary transplants, equal numbers of whole bone marrow cells from leukemic mice were transplanted. We have added more details in the figure legend for Figures 4B and S4D. We agree with the reviewer that the survival difference between WT and KO mice could be attributed to the difference in number of L-GMPs or LSC transplanted in secondary transplants, where we have measurements for the frequency of L-GMPs and LSC. The following sentence has been added into the "Discussion".

Figure legend for Figure 4B

Kaplan Meyer survival curve of sublethally irradiated congenic mice transplanted with 200,000 cells from (left panel) the first replate (right panel) and whole bone marrow cells isolated from leukemic recipients following the first transplant.

Figure legend for Supplemental Figure 4D

Kaplan Meyer survival curve of sublethally irradiated congenic mice transplanted with 200,000 cells from (left panel) the first plate and (right panel) whole bone marrow cells isolated from leukemic recipients following the first transplant.

Line 242

“The prolonged survival in the recipients of MLL-AF6 AML *Sharp1*^{-/-} in the secondary transplants could be explained by lower numbers of transplanted LSC.

Comment 12. The authors mention the cytoplasmic function of AF6 in their introduction but never return to this issue experimentally or in the discussion. Given this, perhaps this is not an important point to make in the introduction?

Response: We agree with the reviewer. The following sentences were removed.

Removed sentences (originally described in Page 4)

“Recent studies highlighted the functions of AF6 in the nucleus, which demonstrated that AF6 functions as a scaffold protein for dimerization of MLL-AF6 and activation of its transcriptional activity¹¹. In addition, AF6 activates rat sarcoma viral oncogene (RAS) downstream targets by regulating RAS-guanosine triphosphate (GTP) levels in the nucleus as a consequence of AF6 shuttling between nucleus and cytoplasm¹³.”

REVIEWERS' COMMENTS:

Reviewer #1 (Remarks to the Author):

The revised manuscript by Numata et al., has added substantial amount of new data and discussion that have now addressed my major remaining concerns. Specifically, I am pleased to see consistent results obtained with 3 different cell lines using 2 different shRNAs, in which one of shRNAs and cell lines were the same as the ones used in Coenen et al., paper where they also saw some degree of growth inhibition. I agree that one possible explanation for the difference was the different degree of KD in particular with the second shRNA only used in Numata's studies. But it is still difficult to explain the result using same shRNA with the same cell line. Since we know that cell lines kept in incubation for years can acquire any mutation, it is possible that some changes may have taken place in the cell lines making them resistant to SHARP1 KD. It may be worth to mention this possibility. Secondly, the authors now also mapped the interaction domain between SHARP1 and MLL-AF6 to MLL 541-1251aa, which is a bit surprising given that SHARP1 is only required for MLL-AF6 but not all MLL fusions transformed cells. But the authors did provide a possible explanation by pointing out unique expression of SHARP1 only in MLL-AF6 transformed cells. Nevertheless, this will have different implications to their working hypothesis/model. I am pleased to see the schematic diagram depicting their findings in Figure 8 has also been updated. But I will suggest to modify or remove the first half of the revised figure legend "Unlike other MLL fusion proteins, MLL-AF6 protein binds to and activates SHARP1 gene in a DOT1L-dependent manner" as it is clear that other MLL fusions should also be able to bind to SHARP1 based on their new data. Overall, it is a much improved and strong manuscript for publication.

Reviewer #2 (Remarks to the Author):

The authors have responded to all my originally comments adequately. I recommend publication of this manuscript.

Reply to the Referees

We thank the referees for their comments. We have made editorial changes as requested. The following is our detailed point-to-point response. We hope that our revised manuscript will be acceptable for publication.

Reviewer #1 (Remarks to the Author):

Comment 1. The revised manuscript by Numata et al., has added substantial amount of new data and discussion that have now addressed my major remaining concerns. Specifically, I am pleased to see consistent results obtained with 3 different cell lines using 2 different shRNAs, in which one of shRNAs and cell lines were the same as the ones used in Coenen et al., paper where they also saw some degree of growth inhibition. I agree that one possible explanation for the difference was the different degree of KD in particular with the second shRNA only used in Numata's studies. But it is still difficult to explain the result using same shRNA with the same cell line. Since we know that cell lines kept in incubation for years can acquire any mutation, it is possible that some changes may have taken place in the cell lines making them resistant to SHARP1 KD. It may be worth to mention this possibility.

Response:

As suggested, we described this possibility in our manuscript.

Line 404, Discussion:

"Also, cell lines may acquire mutations that alter original characteristics after long periods of culture, which could explain differences in knockdown between these two studies."

Comment 2. Secondly, the authors now also mapped the interaction domain between SHARP1 and MLL-AF6 to MLL 541-1251aa, which is a bit surprising given that SHARP1 is only required for MLL-AF6 but not all MLL fusions transformed cells. But the authors did provide a possible explanation by pointing out unique expression of SHARP1 only in MLL-AF6 transformed cells. Nevertheless, this will have different implications to their working hypothesis/model. I am pleased to see the schematic diagram depicting their findings in Figure 8 has also been updated. But I will suggest to modify or remove the first half of the revised figure legend Unlike other MLL fusion proteins, MLL-AF6 protein binds to and activates SHARP1 gene in a DOT1L-dependent manner as it is clear that other MLL fusions should also be able to bind to SHARP1 based on their new data. Overall, it is a much improved and strong manuscript for publication.

Response:

As suggested, we removed the first half of the first sentence in our figure legend for Figure 8. The following is our revised legend for Figure 8:

"MLL-AF6 protein binds to and activates *SHARP1* gene in a DOT1L-dependent manner. Upregulated SHARP1 binds to E-box motifs in active chromatin, and also interacts with MLL-AF6 to regulate a subset of genes critical for leukemogenicity. This unique transcriptional machinery contributes to the maintenance of MLL-AF6 AML leukemic stem cells."

Reviewer #2 (Remarks to the Author):

The authors have responded to all my originally comments adequately. I recommend publication of this manuscript.